# Active Learning for Level Set Estimation Using Randomized Straddle Algorithms

**Yu Inatsu**                                                                          *inatsu.yu@nitech.ac.jp*
*Department of Computer Science, Nagoya Institute of Technology*

**Shion Takeno**                                              *takeno.shion.m6@f.mail.nagoya-u.ac.jp*
*Department of Mechanical Systems Engineering, Graduate School of Engineering, Nagoya University*
*RIKEN Center for Advanced Intelligence Project*

**Kentaro Kutsukake**                                  *kutsukake.kentaro.c3@f.mail.nagoya-u.ac.jp*
*Institute of Materials and Systems for Sustainability, Nagoya University*
*Department of Materials Process Engineering, Graduate School of Engineering, Nagoya University*

**Ichiro Takeuchi**                                        *takeuchi.ichiro.n6@f.mail.nagoya-u.ac.jp*
*Department of Mechanical Systems Engineering, Graduate School of Engineering, Nagoya University*
*RIKEN Center for Advanced Intelligence Project*

**Reviewed on OpenReview:** *https://openreview.net/forum?id=N8M2yqRicS*

## Abstract

Level set estimation (LSE) the problem of identifying the set of input points where a function takes a value above (or below) a given threshold is important in practical applications. When the function is expensive to evaluate and black-box, the straddle algorithm, a representative heuristic for LSE based on Gaussian process models, and its extensions with theoretical guarantees have been developed. However, many existing methods include a confidence parameter, $\beta_t^{1/2}$, that must be specified by the user. Methods that choose $\beta_t^{1/2}$ heuristically do not provide theoretical guarantees. In contrast, theoretically guaranteed values of $\beta_t^{1/2}$ need to be increased depending on the number of iterations and candidate points; they are conservative and do not perform well in practice. In this study, we propose a novel method, the randomized straddle algorithm, in which $\beta_t$ in the straddle algorithm is replaced by a random sample from the chi-squared distribution with two degrees of freedom. The confidence parameter in the proposed method does not require adjustment, does not depend on the number of iterations and candidate points, and is not conservative. Furthermore, we show that the proposed method has theoretical guarantees that depend on the sample complexity and the number of iterations. Finally, we validate the applicability of the proposed method through numerical experiments using synthetic and real data.

## 1 Introduction

In various practical applications, including engineering, level set estimation (LSE) the estimation of the region where the value of a function is above (or below) a given threshold, $\theta$ is important. A specific example of LSE is the estimation of defective regions in materials for quality control. For instance, in silicon ingots, which are used in solar cells, the carrier lifetime value a measure of the ingot's quality is observed at each point on the ingot's surface before shipping, allowing identification of regions that can or cannot be used as solar cells. Since many functions encountered in practical applications, such as the carrier lifetime in the silicon ingot example, are black-box functions with high evaluation costs, it is desirable to identify the desired region without performing an exhaustive search of these black-box functions.

Bayesian optimization (BO) (Shahriari et al., 2015) is a powerful tool for optimizing black-box functions with high evaluation costs. BO predicts black-box functions using surrogate models and adaptively observes the function values based on a criterion called acquisition functions (AFs). Many studies have focused on BO, particularly on developing new AFs. Among these, BO based on the AF known as Gaussian process upper confidence bound (GP-UCB) (Srinivas et al., 2010) offers a theoretical guarantee for finding the optimal solution and is a useful method that is flexible and extendable to various problem settings. GP-UCB-based methods have been proposed in various settings, such as the LSE algorithm (Gotovos et al., 2013), multi-fidelity BO (Kandasamy et al., 2016; 2017), multi-objective BO (Zuluaga et al., 2016; Inatsu et al., 2024), high-dimensional BO (Kandasamy et al., 2015; Rolland et al., 2018), parallel BO (Contal et al., 2013), cascade BO (Kusakawa et al., 2022), and robust BO (Kirschner et al., 2020). These GP-UCB-based methods, like the original GP-UCB-based BO, provide some theoretical guarantee for optimality in each problem setting.

However, GP-UCB and its related methods require the user to specify a confidence parameter, $\beta_t^{1/2}$, to adjust the trade-off between exploration and exploitation, where $t$ is the number of iterations in BO. As a theoretical value for GP-UCB, Srinivas et al. (2010) proposes that $\beta_t^{1/2}$ should increase with the iteration $t$, but this value is conservative, and Takeno et al. (2023) has pointed out that it results in poor practical performance. To solve this issue, Berk et al. (2021) proposed a randomized GP-UCB (RGP-UCB), replacing $\beta_t$ with a ramdom sample from a gamma distribution. They demonstarated through numerical experiments that the practical performance of RGP-UCB is better than that of the original GP-UCB. Althought they also provided the theoretical regret analysis, their theoretical results and proofs of theorems contain some non-ignorable technical issues (see, Appendix C in Takeno et al. (2023)). Recently, Takeno et al. (2023) proposed an improved RGP-UCB (IRGP-UCB), which uses an AF that randomizes $\beta_t$ in GP-UCB by replacing it with a random sample from a two-parameter exponential distribution. IRGP-UCB does not require parameter tuning, and the realized values from the exponential distribution are less conservative than the theoretical values in GP-UCB, resulting in better practical performance. Furthermore, it has been shown that IRGP-UCB provides a tighter bound for the Bayesian regret, one of the optimality measures in BO, than existing methods. However, it is not clear whether IRGP-UCB can be extended to various methods, including LSE. This study proposes a new method for LSE based on the randomization used in IRGP-UCB.

## 1.1 Related Work

GPs (Rasmussen & Williams, 2005) are often used as surrogate models in BO[1], and methods using GPs for LSE have also been proposed. A representative heuristic using GPs is the straddle heuristic by Bryan et al. (2005). The straddle method balances the trade-off between the absolute value of the difference between the GP model's predicted mean and the threshold value, and the uncertainty of the prediction. However, no theoretical analysis has been performed on this method. An extension of the straddle heuristic to cases where the black-box function is a composite function was proposed by Bryan & Schneider (2008), but this too is a heuristic method that lacks theoretical analysis.

As a GP-UCB-based method using GPs, Gotovos et al. (2013) proposed the LSE algorithm. The LSE algorithm uses the same confidence parameter, $\beta_t^{1/2}$ as GP-UCB and is based on the degree of violation from the threshold relative to the confidence interval determined by the GP prediction model. It has been shown that the LSE algorithm returns an $\epsilon$-accurate solution for the true set with high probability. Bogunovic et al. (2016) proposed the truncated variance reduction (TRUVAR) method, which can handle both BO and LSE. TRUVAR also accounts for situations where the observation cost varies across observation points and is designed to maximize the reduction in uncertainty in the uncertain set for each observation point per unit cost. Additionally, Shekhar & Javidi (2019) proposed a chaining-based method, which handles the case where the input space is continuous. As an expected improvement-based method, Zanette et al. (2019) proposed the maximum improvement for level-set estimation (MILE) method. MILE is an algorithm that selects the input point with the highest expected number of points estimated to be in the super-level set, one step ahead, based on data observation.

---

[1] Although both BO and LSE for black-box functions adaptively select the next input point, their objectives are fundamentally different and we will leave the detailed description of BO to a comprehensive survey (Shahriari et al., 2015).

LSE methods have also been proposed for different settings of black-box functions. For example, Letham et al. (2022) introduced a method for cases where the observation of the black-box function is binary. In the robust BO framework, where the inputs of black-box functions are subject to uncertainty, LSE methods for various robust measures have been developed. Iwazaki et al. (2020) proposed LSE for probability threshold robustness measures, and Inatsu et al. (2021) introduced LSE for distributionally robust probability threshold robustness measures both of which are acquisition functions based on MILE. Additionally, Hozumi et al. (2023) proposed a straddle-based method within the framework of transfer learning, where a large amount of data for similar functions is available alongside the primary black-box function to be classified. Inatsu et al. (2020) introduced a MILE-based method for the LSE problem in settings where the uncertainty of the input changes depending on the cost. Mason et al. (2022) addressed the LSE problem in the context where the black-box function is an element of a reproducing kernel Hilbert space.

The straddle method, LSE algorithm, TRUVAR, chaining-based algorithm, and MILE, which have been proposed under settings similar to those considered in this study, have the following issues. The straddle method is not an acquisition function proposed based on GP-UCB, but it includes the confidence parameter $\beta_t^{1/2}$, which is essentially the same as in GP-UCB. However, the value of this parameter is determined heuristically, resulting in a method without theoretical guarantees. The LSE algorithm and TRUVAR have been theoretically analyzed, but, like GP-UCB, they require increasing the theoretical value of the confidence parameter according to the iteration $t$, which makes them conservative. The chaining-based algorithm can handle continuous spaces through discretization, but it involves many adjustment parameters. The recommended theoretical values depend on model parameters, including kernel parameters of the surrogate model, and are known only for specific settings. MILE is designed for cases with a finite number of candidate points and does not support continuous settings like the chaining-based algorithm.

## 1.2 Contribution

This study proposes a novel straddle AF called the *randomized straddle*, which introduces the confidence parameter randomization technique used in IRGP-UCB and solves the problems described in Section 1.1. Figure 1 shows a comparison of the confidence parameters in the proposed AF and those in the LSE algorithm. The contributions of this study are as:

- This study proposes a randomized straddle AF, which replaces $\beta_t$ in the straddle heuristic with a random sample from the chi-squared distribution with two degrees of freedom (one-parameter exponential distribution with parameter $1/2$). We emphasize that unlike the LSE algorithm, the confidence parameter in the randomized straddle does not need to increase with the iteration $t$. Additionally, $\beta_t^{1/2}$ in the LSE algorithm depends on the number of candidate points $|\mathcal{X}|$, and $\beta_t^{1/2}$ increases as $|\mathcal{X}|$ increases, while $\beta_t^{1/2}$ in the randomized straddle does not depend on $|\mathcal{X}|$, and can be applied even when $\mathcal{X}$ is an infinite set. Furthermore, the expected value of the realized value of $\beta_t^{1/2}$ in the randomized straddle is $\sqrt{2\pi}/2 \approx 1.25$, which is less conservative than the theoretical value in the LSE algorithm.

- We show that the randomized straddle guarantees that the expected loss for misclassification in LSE converges to 0. In particular, for the misclassification loss $r_t = \frac{1}{|\mathcal{X}|} \sum_{\boldsymbol{x} \in \mathcal{X}} l_t(\boldsymbol{x})$, the randomized straddle guarantees $\mathbb{E}[r_t] = O(\sqrt{\gamma_t/t})$, where $l_t(\boldsymbol{x})$ is 0 if the input point $\boldsymbol{x}$ is correctly classified, and $|f(\boldsymbol{x}) - \theta|$, if misclassified, and $\gamma_t$ is the maximum information gain which is a commonly used sample complexity measure.

- Additionally, we conducted numerical experiments using synthetic and real data, which confirmed that the proposed method has performance equal to or better than existing methods.

## 2 Preliminary

Let $f : \mathcal{X} \to \mathbb{R}$ be an expensive-to-evaluate black-box function, where $\mathcal{X} \subset \mathbb{R}^d$ is a finite set, or an infinite compact set with positive Lebesgue measure $\mathrm{Vol}(\mathcal{X})$. Also let $\theta \in \mathbb{R}$ be a known threshold given by the user.

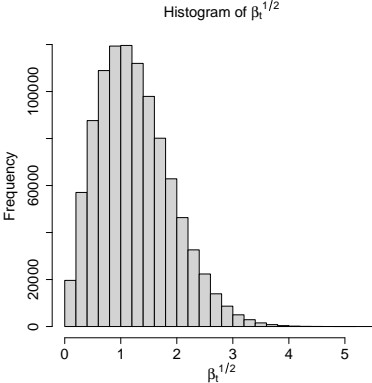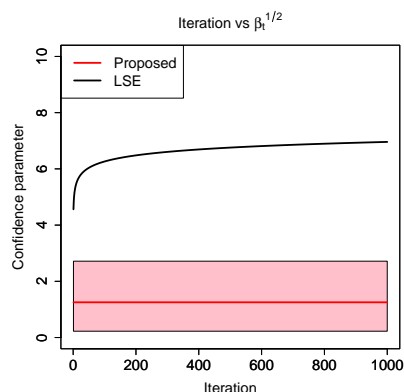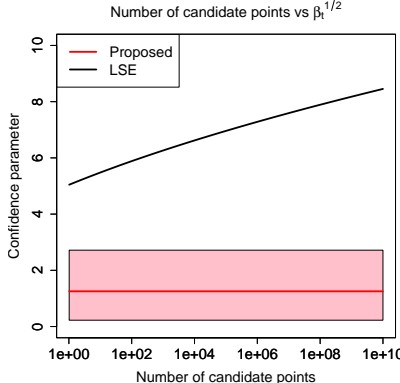

Figure 1: Comparison of the confidence parameter $\beta_t^{1/2}$ in the randomized straddle and LSE algorithms. The left-hand side figure shows the histogram of $\beta_t^{1/2}$ when $\beta_t$ is sampled 1,000,000 times from the chi-squared distribution with two degrees of freedom. The red line in the center and right figure denotes $\mathbb{E}[\beta_t^{1/2}] = \sqrt{2\pi}/2 \approx 1.25$, the shaded area denotes the 95% confidence interval of $\beta_t^{1/2}$, and the black line denotes the theoretical value of $\beta_t^{1/2}$ in the LSE algorithm given by $\beta_t^{1/2} = \sqrt{2\log(|\mathcal{X}|\pi^2 t^2/(6\delta))}$, where $\delta = 0.05$. The figure in the center shows the behavior of $\beta_t^{1/2}$ as the number of iterations $t$ increases when the number of candidate points $|\mathcal{X}|$ is fixed at 1000, whereas the figure on the right shows the behavior of $\beta_t^{1/2}$ as the number of candidate points $|\mathcal{X}|$ increases when the number of iterations $t$ is fixed at 100.

The aim of this study is to efficiently identify subsets $H^*$ and $L^*$ of $\mathcal{X}$ defined as

$$H^* = \{\boldsymbol{x} \in \mathcal{X} \mid f(\boldsymbol{x}) \geq \theta\}, \ L^* = \{\boldsymbol{x} \in \mathcal{X} \mid f(\boldsymbol{x}) < \theta\}.$$

For each iteration $t \geq 1$, we can query $\boldsymbol{x}_t \in \mathcal{X}$, and $f(\boldsymbol{x}_t)$ is observed with noise as $y_t = f(\boldsymbol{x}_t) + \varepsilon_t$, where $\varepsilon_t$ follows the normal distribution with mean 0 and variance $\sigma_{\mathrm{noise}}^2$. In this study, we assume that $f$ is a sample path from a GP $\mathcal{GP}(0, k)$, where $\mathcal{GP}(0, k)$ is the zero mean GP with a kernel function $k(\cdot, \cdot)$. Moreover, we assume that $k(\cdot, \cdot)$ is a positive-definite kernel that satisfies $k(\boldsymbol{x}, \boldsymbol{x}) \leq 1$ for all $\boldsymbol{x} \in \mathcal{X}$, and $f, \varepsilon_1, \ldots, \varepsilon_t$ are mutually independent.

**Gaussian Process Model** We use a GP surrogate model $\mathcal{GP}(0, k)$ for the black-box function. Given a dataset $D_t = \{(\boldsymbol{x}_j, y_j)\}_{j=1}^t$, where $t \geq 1$ is the number of iterations, the posterior distribution of $f$ is again a GP. Then, its posterior mean $\mu_t(\boldsymbol{x})$ and posterior variance $\sigma_t^2(\boldsymbol{x})$ can be calculated as:

$$
\begin{aligned}
\mu_t(\boldsymbol{x}) &= \boldsymbol{k}_t(\boldsymbol{x})^\top (\boldsymbol{K}_t + \sigma_{\mathrm{noise}}^2 \boldsymbol{I}_t)^{-1} \boldsymbol{y}_t, \\
\sigma_t^2(\boldsymbol{x}) &= k(\boldsymbol{x}, \boldsymbol{x}) - \boldsymbol{k}_t(\boldsymbol{x})^\top (\boldsymbol{K}_t + \sigma_{\mathrm{noise}}^2 \boldsymbol{I}_t)^{-1} \boldsymbol{k}_t(\boldsymbol{x}),
\end{aligned}
\tag{1}
$$

where $\boldsymbol{k}_t(\boldsymbol{x})$ is the $t$-dimensional vector whose $i$-th element is $k(\boldsymbol{x}, \boldsymbol{x}_i)$, $\boldsymbol{y}_t = (y_1, \ldots, y_t)^\top$, $\boldsymbol{K}_t$ is the $t \times t$ matrix whose $(j, k)$-th element is $k(\boldsymbol{x}_j, \boldsymbol{x}_k)$, $\boldsymbol{I}_t$ is the $t \times t$ identity matrix, with a superscript $\top$ that indicates the transpose of vectors or matrices. In addition, we define $D_0 = \emptyset$, $\mu_0(\boldsymbol{x}) = 0$ and $\sigma_0^2(\boldsymbol{x}) = k(\boldsymbol{x}, \boldsymbol{x})$.

# 3 Proposed Method

In this section, we describe a method for estimating $H^*$ and $L^*$ based on the GP posterior and an AF for determining the next evaluation.

### 3.1 Level Set Estimation

First, we propose a method to estimate $H^*$ and $L^*$. While an existing study (Gotovos et al., 2013) proposes an estimation method using the lower and upper bounds of a credible interval of $f(\boldsymbol{x})$, this study proposes an estimation method using the posterior mean instead of the credible interval.

**Definition 3.1** (Level Set Estimation)**.** For each $t \geq 1$, we estimate $H^*$ and $L^*$ as:

$$H_t = \{\boldsymbol{x} \in \mathcal{X} \mid \mu_{t-1}(\boldsymbol{x}) \geq \theta\}, \ L_t = \{\boldsymbol{x} \in \mathcal{X} \mid \mu_{t-1}(\boldsymbol{x}) < \theta\}. \tag{2}$$

By definition 3.1, any $\boldsymbol{x} \in \mathcal{X}$ belongs to either $H_t$ or $L_t$, and $H_t \cup L_t = \mathcal{X}$. Therefore, the unknown set, as in existing study (Gotovos et al., 2013), is not defined in this study.

### 3.2 Acquisition Function

In this section, we propose an AF for determining the next point to be evaluated. For each $t \geq 1$ and $\boldsymbol{x} \in \mathcal{X}$, we define the upper bound $\mathrm{ucb}_{t-1}(\boldsymbol{x})$ and lower bound $\mathrm{lcb}_{t-1}(\boldsymbol{x})$ in the credible interval of $f(\boldsymbol{x})$ as

$$\mathrm{ucb}_{t-1}(\boldsymbol{x}) = \mu_{t-1}(\boldsymbol{x}) + \beta_t^{1/2}\sigma_{t-1}(\boldsymbol{x}), \ \mathrm{lcb}_{t-1}(\boldsymbol{x}) = \mu_{t-1}(\boldsymbol{x}) - \beta_t^{1/2}\sigma_{t-1}(\boldsymbol{x}),$$

where $\beta_t^{1/2} \geq 0$ is a user-specified confidence parameter. Here, the straddle heuristic $\mathrm{STR}_{t-1}(\boldsymbol{x})$ proposed by Bryan et al. (2005) is defined as:

$$\mathrm{STR}_{t-1}(\boldsymbol{x}) = \beta_t^{1/2}\sigma_{t-1}(\boldsymbol{x}) - |\mu_{t-1}(\boldsymbol{x}) - \theta|.$$

Thus, by using $\mathrm{ucb}_{t-1}(\boldsymbol{x})$ and $\mathrm{lcb}_{t-1}(\boldsymbol{x})$, $\mathrm{STR}_{t-1}(\boldsymbol{x})$ can be rewritten as

$$\mathrm{STR}_{t-1}(\boldsymbol{x}) = \min\{\mathrm{ucb}_{t-1}(\boldsymbol{x}) - \theta, \theta - \mathrm{lcb}_{t-1}(\boldsymbol{x})\}.$$

We consider sampling $\beta_t$ of the straddle heuristic from a probability distribution. In the framework of black-box function maximization, Takeno et al. (2023) uses a sample from a two-parameter exponential distribution as the confidence parameter of the original GP-UCB. The two-parameter exponential distribution considered by Takeno et al. (2023) can be expressed as $2\log(|\mathcal{X}|/2) + s_t$, where $s_t$ follows the chi-squared distribution with two degrees of freedom. Therefore, we use a similar argument and consider $\beta_t$ of the straddle heuristic as a sample from the chi-squared distribution with two degrees of freedom, and propose the following randomized straddle AF.

**Definition 3.2** (Randomized Straddle)**.** For each $t \geq 1$, let $\beta_t$ be a sample from the chi-squared distribution with two degrees of freedom, where $\beta_1, \ldots, \beta_t, \varepsilon_1, \ldots, \varepsilon_t, f$ are mutually independent. Then, the randomized straddle $a_{t-1}(\boldsymbol{x})$ is defined as follows:

$$a_{t-1}(\boldsymbol{x}) = \max\{\min\{\mathrm{ucb}_{t-1}(\boldsymbol{x}) - \theta, \theta - \mathrm{lcb}_{t-1}(\boldsymbol{x})\}, 0\}. \tag{3}$$

Hence, using $a_{t-1}(\boldsymbol{x})$, the next point to be evaluated is selected by $\boldsymbol{x}_t = \arg\max_{\boldsymbol{x} \in \mathcal{X}} a_{t-1}(\boldsymbol{x})$. Figure 2 shows the difference in the input points selected when using $a_{t-1}(\boldsymbol{x})$ with different $\beta_t^{1/2}$. Takeno et al. (2023) adds a constant $2\log(|\mathcal{X}|/2)$, which depends on the number of elements in $\mathcal{X}$, to the sample from the chi-squared distribution with two degrees of freedom. In contrast, the random sample proposed in this study does not require the addition of such a constant. As a result, the confidence parameter in the randomized straddle does not depend on the number of iterations $t$ or the number of candidate points. The only difference between the straddle heuristic $\mathrm{STR}_{t-1}(\boldsymbol{x})$ and equation 3 is that $\beta_t^{1/2}$ is randomized, and equation 3 performs a max operation with 0. We describe in Section 4 that this modification leads to theoretical guarantees. Finally, we give the pseudocode of the proposed algorithm in Algorithm 1.

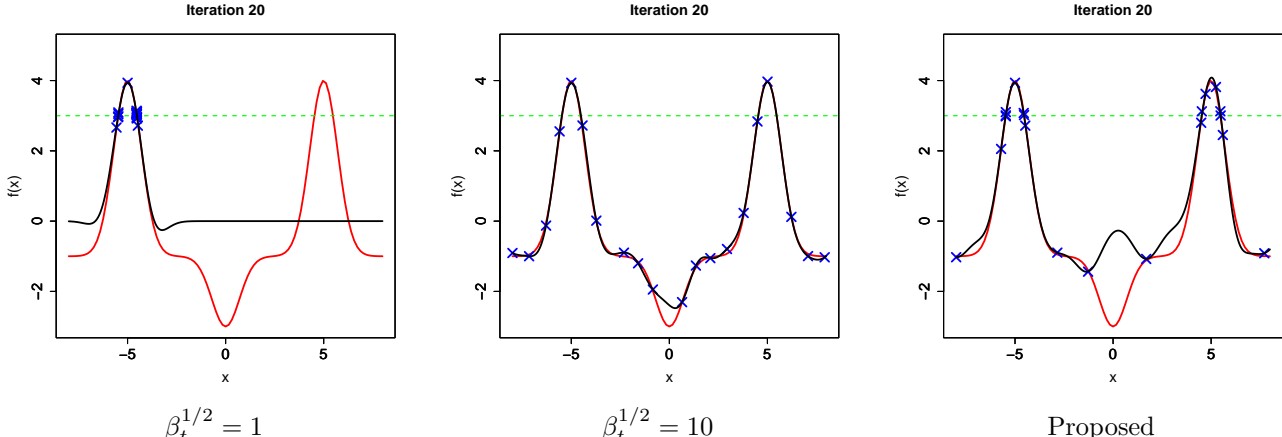

$$\beta_t^{1/2} = 1 \qquad\qquad \beta_t^{1/2} = 10 \qquad\qquad \text{Proposed}$$

Figure 2: Comparison of points selected by $a_{t-1}(x)$ with different $\beta_t^{1/2}$. The red line represents the true black-box function $f(x) = 5\exp(-(x+5)^2) + 5\exp(-(x-5)^2) - 2\exp(-x^2) - 1$, the black line represents the posterior mean, and the blue crosses represent the observed points. The figures on the left, center and right show the differences for 20 observation points when using, respectively, $\beta_t^{1/2} = 1$, $\beta_t^{1/2} = 10$ and $\beta_t$ which follows the chi-squared distribution with two degrees of freedom, in the calculation of $a_{t-1}(x)$, where $x = -5$ is chosen as the initial point under the observation noise $\sigma_{\text{noise}}^2 = 10^{-2}$ and threshold $\theta = 3$. Since $\text{STR}_{t-1}(x)$ is represented as $\text{STR}_{t-1}(x) = \beta_t^{1/2}\sigma_{t-1}(x) - |\mu_{t-1}(x) - \theta|$, when $\beta_t^{1/2} = 1$, $\beta_t^{1/2}$ is small so the second term of $\text{STR}_{t-1}(x)$ dominates, and as a result, it can be seen that only values whose posterior mean are close to the threshold are observed. Conversely, when $\beta_t^{1/2} = 10$, $\beta_t^{1/2}$ is large so the first term of $\text{STR}_{t-1}(x)$ dominates, resulting in the AF that is almost the same as uncertainty sampling, and it can be seen that the selected inputs are spaced almost equally apart. On the other hand, these behaviors are not observed with the proposed method.

---

**Algorithm 1** Active Learning for Level Set Estimation Using Randomized Straddle Algorithms

---

**Input:** GP prior $\mathcal{GP}(0, k)$, threshold $\theta \in \mathbb{R}$
  **for** $t = 1, 2, \ldots, T$ **do**
    Compute $\mu_{t-1}(\boldsymbol{x})$ and $\sigma_{t-1}^2(\boldsymbol{x})$ for each $\boldsymbol{x} \in \mathcal{X}$ by equation 1
    Estimate $H_t$ and $L_t$ by equation 2
    Generate $\beta_t$ from the chi-squared distribution with two degrees of freedom
    Compute $\text{ucb}_{t-1}(\boldsymbol{x})$, $\text{lcb}_{t-1}(\boldsymbol{x})$ and $a_{t-1}(\boldsymbol{x})$
    Select the next evaluation point $\boldsymbol{x}_t$ by $\boldsymbol{x}_t = \arg\max_{\boldsymbol{x} \in \mathcal{X}} a_{t-1}(\boldsymbol{x})$
    Observe $y_t = f(\boldsymbol{x}_t) + \varepsilon_t$ at the point $\boldsymbol{x}_t$
    Update GP by adding the observed data
  **end for**
**Output:** Return $H_T$ and $L_T$ as the estimated sets

---

## 4 Theoretical Analysis

In this section, we give theoretical guarantees for the proposed model. First, we define the loss $l_t(\boldsymbol{x})$ for each $\boldsymbol{x} \in \mathcal{X}$ and $t \geq 1$ as

$$l_t(\boldsymbol{x}) = \begin{cases} 0 & \text{if } \boldsymbol{x} \in H^*, \boldsymbol{x} \in H_t, \\ 0 & \text{if } \boldsymbol{x} \in L^*, \boldsymbol{x} \in L_t, \\ f(\boldsymbol{x}) - \theta & \text{if } \boldsymbol{x} \in H^*, \boldsymbol{x} \in L_t, \\ \theta - f(\boldsymbol{x}) & \text{if } \boldsymbol{x} \in L^*, \boldsymbol{x} \in H_t \end{cases} .$$

Then, the loss $r(H_t, L_t)$ for the estimated sets $H_t$ and $L_t$ is defined as [2]:

$$r(H_t, L_t) = \begin{cases} \frac{1}{|\mathcal{X}|} \sum_{\boldsymbol{x} \in \mathcal{X}} l_t(\boldsymbol{x}) & \text{if } \mathcal{X} \text{ is finite} \\ \frac{1}{\text{Vol}(\mathcal{X})} \int_{\mathcal{X}} l_t(\boldsymbol{x}) \mathrm{d}\boldsymbol{x} & \text{if } \mathcal{X} \text{ is infinite} \end{cases}$$
$$\equiv r_t.$$

We also define the cumulative loss as $R_t = \sum_{i=1}^t r_i$. Let $\gamma_t$ be a maximum information gain[3], where $\gamma_t$ is one of indicators for measuring the sample complexity. The maximum information gain $\gamma_t$ is often used in theoretical analysis of BO and LSE using GP (Srinivas et al., 2010; Gotovos et al., 2013), and $\gamma_t$ is given by

$$\gamma_t = \frac{1}{2} \sup_{\{\tilde{\boldsymbol{x}}_1, \dots, \tilde{\boldsymbol{x}}_t\} \subset \mathcal{X}} \log \det(\boldsymbol{I}_t + \sigma_{\text{noise}}^{-2} \tilde{\boldsymbol{K}}_t), \tag{4}$$

where $\tilde{\boldsymbol{x}}_1, \dots, \tilde{\boldsymbol{x}}_t$ are any elements of $\mathcal{X}$, and $\tilde{\boldsymbol{K}}_t$ is the $t \times t$ matrix whose $(j,k)$-th element is $k(\tilde{\boldsymbol{x}}_j, \tilde{\boldsymbol{x}}_k)$. Then, the following theorem holds.

**Theorem 4.1.** Assume that $f$ follows $\mathcal{GP}(0, k)$, where $k(\cdot, \cdot)$ is a positive-definite kernel satisfying $k(\boldsymbol{x}, \boldsymbol{x}) \leq 1$ for any $\boldsymbol{x} \in \mathcal{X}$. For each $t \geq 1$, let $\beta_t$ be a sample from the chi-squared distribution with two degrees of freedom, where $\beta_1, \dots, \beta_t, \varepsilon_1, \dots \varepsilon_t, f$ are mutually independent. Then, the following inequality holds:

$$\mathbb{E}[R_t] \leq \sqrt{C_1 t \gamma_t},$$

where $C_1 = 4/\log(1 + \sigma_{\text{noise}}^{-2})$, and the expectation is taken with all randomness including $f$, $\varepsilon_t$ and $\beta_t$.

From Theorem 4.1, the following theorem holds.

**Theorem 4.2.** Under the assumptions of Theorem 4.1, the following inequality holds:

$$\mathbb{E}[r_t] \leq \sqrt{\frac{C_1 \gamma_t}{t}},$$

where $C_1$ is given in Theorem 4.1.

Note that Theorem 4.1 and 4.2 hold whether $\mathcal{X}$ is a finite or infinite set. By the definition of the loss $l_t(\boldsymbol{x})$, $l_t(\boldsymbol{x})$ represents how far $f(\boldsymbol{x})$ is from the threshold when $\boldsymbol{x}$ is misclassified, and $r_t$ represents the average value of $l_t(\boldsymbol{x})$ across all candidate points. Under mild assumptions, it is known that $\gamma_t$ is sublinear (Srinivas et al., 2010). Therefore, by Theorem 4.1, it is guaranteed that $R_t$ is also sublinear in the expected value sense. Furthermore, by Theorem 4.2, it is guaranteed that $r_t$ converges to 0 in the expected value sense. Here, we must emphasize that Theorem 4.1 and 4.2 cannot be derived by simply randomizing the confidence parameters of existing methods. First, although the proposed method is similar to existing methods in that it randomly samples the confidence parameters of the AF, several issues had to be resolved in order to derive theoretical guarantees in the LSE setting addressed in this paper. The proposed method is inspired by IRGP-UCB in Takeno et al. (2023), but they deal with maximization problems in the first place and consider regret $f(\boldsymbol{x}^*) - f(\boldsymbol{x}_t)$, which is the difference between the maximum value $f(\boldsymbol{x}^*)$ and the function value $f(\boldsymbol{x}_t)$ at the observation point $\boldsymbol{x}_t$. They consider a Bayesian cumulative (or simple) regret as an evaluation index for theoretical analysis, and the theoretical validity of their method is based on the fact that $f(\boldsymbol{x}^*)$ can be bounded from above with high probability, and as a result, the expected value of $f(\boldsymbol{x}^*)$ can be bounded above by a certain expected value (Lemma 4.1 and 4.2 in Takeno et al. (2023)), which is achieved by using UCB. On the other hand, the losses $l_t(\boldsymbol{x})$ and $r_t(H_t, L_t)$ of the LSE addressed in this paper are essentially different from the regret $f(\boldsymbol{x}^*) - f(\boldsymbol{x}_t)$ and Bayesian cumulative (or simple) regret in maximization problems. Therefore, it was not clear whether claims similar to Lemma 4.1 and 4.2 in Takeno et al. (2023) could be derived by simply randomizing the confidence parameters of some AF of LSE. In addition, in existing LSE

---

[2]The discussion of the case where the loss is defined based on the maximum value $r(H_t, L_t) = \max_{\boldsymbol{x} \in \mathcal{X}} l_t(\boldsymbol{x})$ is given in Appendix A.

[3] According to equation 4, $\gamma_t$ should be called the supremum information gain, but since the term maximum information gain is also used when using a sup operator (see, e.g., Vakili et al. (2023)), in the rest of this paper we will continue to call $\gamma_t$ the maximum information gain.

studies such as Gotovos et al. (2013), the classification rule includes the posterior mean, posterior standard deviation, and $\beta_t$, and the loss function depends on the classification rule. Therefore, since the classification rule includes $\beta_t$, theoretical analysis based on randomization was difficult. On the other hand, although the loss of the proposed method is based on the classification rule, unlike theirs, the classification rule itself does not include $\beta_t$ because the classification rule uses only the posterior mean, and as a result, randomization analysis became possible.

On the other hand, it is challenging to directly compare the proposed method with GP-based methods such as the LSE algorithm and TRUVAR in terms of theoretical analysis. This difficulty arises because, first, the proposed method and these methods use different approaches to estimate $H^*$ and $L^*$, and second, the criteria for evaluating the quality of the estimated sets differ. However, it is important to note that the proposed method has theoretical guarantees, and the confidence parameter $\beta_t^{1/2}$ does not depend on the number of iterations $t$ or the input space $\mathcal{X}$, making it applicable whether $\mathcal{X}$ is finite or infinite. Additionally, since $\mathbb{E}[\beta_t^{1/2}] = \sqrt{2\pi}/2 \approx 1.25$, the realized values of $\beta_t^{1/2}$ are not conservative. To the best of our knowledge, no existing method satisfies all of these properties. Moreover, we confirm in Section 5 that the practical performance of the proposed method is equal to or better than existing methods.

Finally, we give a theorem on high-probability bounds for $R_t$ and $r_t$ when using the proposed method. Since Theorem 4.1 and 4.2 provide bounds on the expected values of $R_t$ and $r_t$, we can easily derive high-probability bounds by using Markov's inequality, $\mathbb{P}(|X| \geq a) \leq \frac{\mathbb{E}[|X|]}{a}$, where $X$ is a random variable and $a$ is a positive number. If $R_t$ is used as $X$, then since $|R_t| = R_t$, using Theorem 4.1 and Markov's inequality the inequality $\mathbb{P}(R_t \geq a) \leq \frac{\sqrt{C_1 t \gamma_t}}{a}$ holds. Therefore, for a given $\delta \in (0,1)$, if we set $a = \delta^{-1}\sqrt{C_1 t \gamma_t}$, then $R_t \leq \delta^{-1}\sqrt{C_1 t \gamma_t}$ holds with probability at least $1 - \delta$. Similarly, for $r_t$, if we set $a = \delta^{-1}\sqrt{C_1 \gamma_t/t}$, then $r_t \leq \delta^{-1}\sqrt{C_1 \gamma_t/t}$ holds with probability at least $1 - \delta$. We summarize these results in Theorem 4.3.

**Theorem 4.3.** Let $\delta \in (0,1)$ and $C_1 = 4/\log(1 + \sigma_{\text{noise}}^{-2})$. For each $t \geq 1$, under the assumptions of Theorem 4.1, the following inequalities hold with probability at least $1 - \delta$:

$$R_t \leq \delta^{-1}\sqrt{C_1 t \gamma_t}, \ r_t \leq \delta^{-1}\sqrt{\frac{C_1 \gamma_t}{t}}.$$

From Theorem 4.3, it is possible to derive high-probability bounds for $R_t$ and $r_t$, but the problem of the right-hand side not being tight remains. Specifically, the term $\delta^{-1}$ remains in the right-hand side. On the other hand, in many studies that deal with high-probability bounds such as Srinivas et al. (2010), the term $\sqrt{\log(\delta^{-1})}$ appears in the bound, which is tighter than $\delta^{-1}$. Therefore, one of the issues for the future is to improve $\delta^{-1}$ in Theorem 4.3 to $\sqrt{\log(\delta^{-1})}$.

## 5 Numerical Experiments

We confirm the practical performance of the proposed method using synthetic functions and real-world data.

### 5.1 Synthetic Data Experiments when $\mathcal{X}$ is Finite

In this section, the input space $\mathcal{X}$ was defined as a set of grid points that uniformly cut the region $[l_1, u_1] \times [l_2, u_2]$ into $50 \times 50$. In all experiments, we used the following Gaussian kernel:

$$k(\boldsymbol{x}, \boldsymbol{x}') = \sigma_f^2 \exp\left(-\frac{\|\boldsymbol{x} - \boldsymbol{x}'\|_2^2}{L}\right).$$

As black-box functions, we considered the following three synthetic functions:

Case 1 The black-box function $f(x_1, x_2)$ is a sample path from $\mathcal{GP}(0, k)$, where $k(\cdot, \cdot)$ is given by

$$k(\boldsymbol{x}, \boldsymbol{x}') = \exp(-\|\boldsymbol{x} - \boldsymbol{x}'\|_2^2/2).$$

Table 1: Experimental parameters for each setting in Section 5.1

| Black-box function | $l_1$ | $u_1$ | $l_2$ | $u_2$ | $\sigma_f^2$ | $L$ | $\sigma_{\text{noise}}^2$ | $\theta$ |
|---|---|---|---|---|---|---|---|---|
| GP sample path | $-5$ | 5 | $-5$ | 5 | 1 | 2 | $10^{-6}$ | 0.5 |
| Sinusoidal function | 0 | 1 | 0 | 2 | $\exp(2)$ | $2\exp(-3)$ | $\exp(-2)$ | 1 |
| Himmelblau's function | -5 | 5 | -5 | 5 | $\exp(8)$ | 2 | $\exp(4)$ | 0 |

**Case 2** The black-box function $f(x_1, x_2)$ is the following sinusoidal function:

$$f(x_1, x_2) = \sin(10x_1) + \cos(4x_2) - \cos(3x_1 x_2).$$

**Case 3** The black-box function $f(x_1, x_2)$ is the following shifted negative Himmelblau function:

$$f(x_1, x_2) = -(x_1^2 + x_2 - 11)^2 - (x_1 + x_2^2 - 7)^2 + 100.$$

Furthermore, we used the normal distribution with mean 0 and variance $\sigma_{\text{noise}}^2$ for the observation noise. The threshold $\theta$ and the parameters used for each setting are summarized in Table 1. The settings for the sinusoidal and Himmelblau functions are the same as those used in Zanette et al. (2019). The performance was evaluated using the loss $r_t$ and $\text{Fscore}_t$, where $\text{Fscore}_t$ is the F-score calculated by

$$\text{Pre}_t = \frac{|H_t \cap H^*|}{|H_t|}, \text{Rec}_t = \frac{|H_t \cap H^*|}{|H^*|}, \text{Fscore}_t = \frac{2 \times \text{Pre}_t \times \text{Rec}_t}{\text{Pre}_t + \text{Rec}_t}.$$

Then, we compared the following six AFs:

(Random) Select $\boldsymbol{x}_t$ by using random sampling.

(US) Perform uncertainty sampling, that is, $\boldsymbol{x}_t = \arg\max_{\boldsymbol{x}\in\mathcal{X}} \sigma_{t-1}^2(\boldsymbol{x})$.

(Straddle) Perform the straddle heuristic proposed by Bryan et al. (2005), that is, $\boldsymbol{x}_t = \arg\max_{\boldsymbol{x}\in\mathcal{X}} \text{STR}_{t-1}(\boldsymbol{x})$.

(LSE) Perform the LSE algorithm using the LSE AF $a_{t-1}^{(\text{LSE})}(\boldsymbol{x})$ proposed by Gotovos et al. (2013), that is, $\boldsymbol{x}_t = \arg\max_{\boldsymbol{x}\in\mathcal{X}} a_{t-1}^{(\text{LSE})}(\boldsymbol{x})$.

(MILE) Perform the MILE algorithm proposed by Zanette et al. (2019), that is, $\boldsymbol{x}_t = \arg\max_{\boldsymbol{x}\in\mathcal{X}} a_{t-1}^{(\text{MILE})}(\boldsymbol{x})$, where, $a_{t-1}^{(\text{MILE})}(\boldsymbol{x})$ is the same as the robust MILE, another AF proposed by Zanette et al. (2019), with the tuning parameters $\epsilon$ and $\gamma$ set to 0 and $-\infty$, respectively.

(Proposed) Select $\boldsymbol{x}_t$ by using equation 3, that is, $\boldsymbol{x}_t = \arg\max_{\boldsymbol{x}\in\mathcal{X}} a_{t-1}(\boldsymbol{x})$.

In all experiments, the classification rules were the same for all six methods, and only the AF was changed. We used $\beta_t^{1/2} = 3$ as the confidence parameter required for MILE and Straddle, and $\beta_t^{1/2} = \sqrt{2\log(2500 \times \pi^2 t^2/(6 \times 0.05))}$ for LSE. Under this setup, one initial point was taken at random and the algorithm was run until the number of iterations reached 300. This simulation was repeated 100 times, and the average $r_t$ and $\text{Fscore}_t$ at each iteration were calculated, where in Case 1, $f$ was generated for each simulation from $\mathcal{GP}(0, k)$.

As shown in Fig. 3, the proposed method consistently performs as well as or better than the comparison methods in all three cases, in terms of both the loss $r_t$ and the $\text{Fscore}_t$.

## 5.2 Synthetic Data Experiments when $\mathcal{X}$ is Infinite

In this section, we used the region $[-5, 5]^5 \subset \mathbb{R}^5$ as $\mathcal{X}$ and the same kernel as in Section 5.1. As black-box functions, we used the following three synthetic functions:

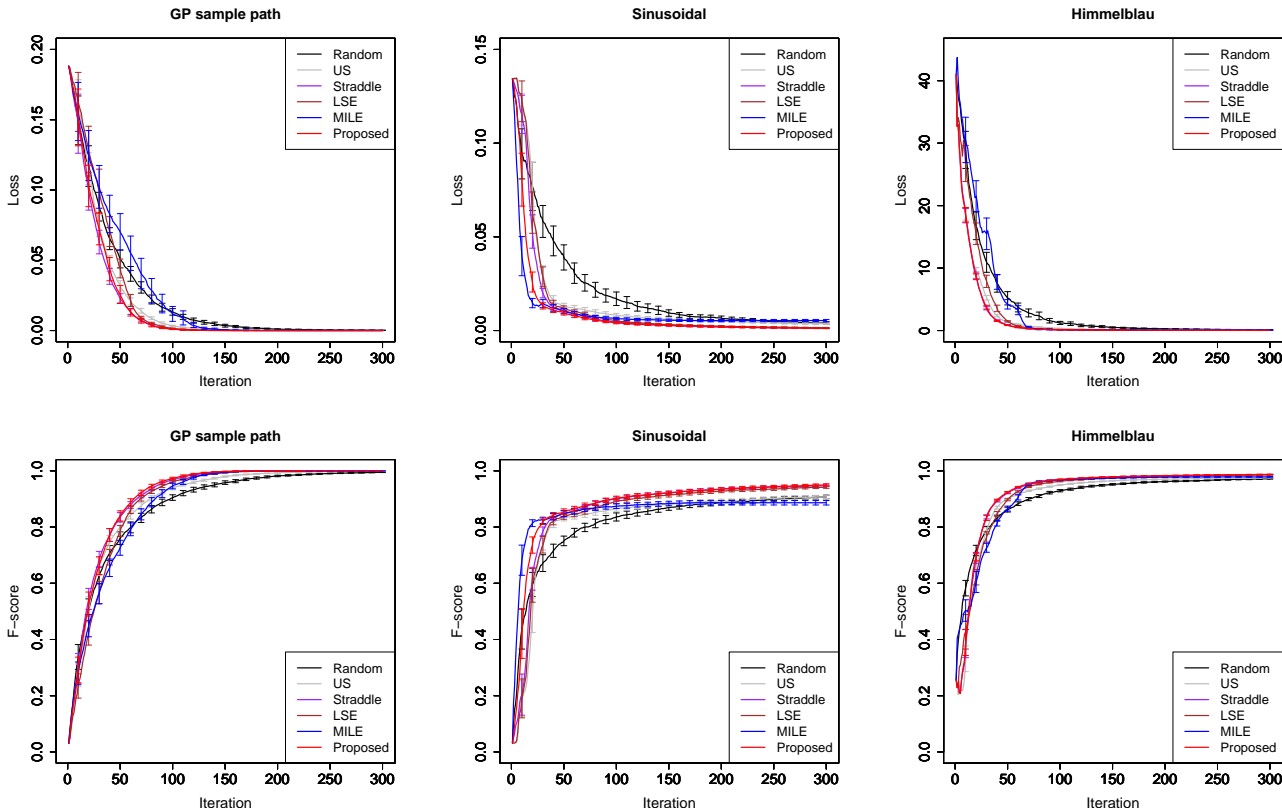

Figure 3: Averages for the loss $r_t$ and Fscore$_t$ for each AF over 100 simulations across different settings when the input space is finite. The top row shows $r_t$, and the bottom row shows Fscore$_t$. Error bars represent six times the standard error.

**Case 1** The black-box function $f(x_1, x_2, x_3, x_4, x_5)$ is the following shifted negative sphere function:

$$f(x_1, x_2, x_3, x_4, x_5) = 41.65518 - \left( \sum_{d=1}^{5} x_d^2 \right).$$

**Case 2** The black-box function $f(x_1, x_2, x_3, x_4, x_5)$ is the following shifted negative Rosenbrock function:

$$f(x_1, x_2, x_3, x_4, x_5) = 53458.91 - \left[ \sum_{d=1}^{4} \left\{ 100(x_{d+1} - x_d^2)^2 + (1 - x_d)^2 \right\} \right].$$

**Case 3** The black-box function $f(x_1, x_2, x_3, x_4, x_5)$ is the following shifted negative Styblinski-Tang function:

$$f(x_1, x_2, x_3, x_4, x_5) = -20.8875 - \frac{\sum_{d=1}^{5}(x_d^4 - 16x_d^2 + 5x_d)}{2}.$$

Additionally, we used the normal distribution with mean 0 and variance $\sigma_{\text{noise}}^2$ for the observation noise. The threshold $\theta$ and parameters used for each setting are summarized in Table 2. The performance was evaluated using $r_t$ and Fscore$_t$. For each simulation, 100,000 points were randomly selected from $[-5, 5]^5$, which were used as the input point set $\tilde{X}$ to calculate $r_t$ and Fscore$_t$. The values of $r_t$ and Fscore$_t$ in $\tilde{X}$ were calculated as approximations of the true values. As AFs, we compared five methods used in Section 5.1, except for MILE, which does not handle continuous settings. We used $\beta_t^{1/2} = 3$ as the confidence parameter required

Table 2: Experimental parameters for each setting in Section 5.2

| Black-box function | $\sigma_f^2$ | $L$ | $\sigma_{\text{noise}}^2$ | $\theta$ |
|---|---|---|---|---|
| Sphere | 900 | 40 | $10^{-6}$ | 9.6 |
| Rosenbrock | $30000^2$ | 40 | $10^{-6}$ | 14800 |
| Styblinski-Tang | $75^2$ | 40 | $10^{-6}$ | 12.3 |

for Straddle, and $\beta_t^{1/2} = \sqrt{2\log(10^{15} \times \pi^2 t^2/(6 \times 0.05))}$ for LSE. Here, the original LSE algorithm uses the intersection of $\text{ucb}_{t-1}(\boldsymbol{x})$ and $\text{lcb}_{t-1}(\boldsymbol{x})$ in the previous iterations given below to calculate the AF:

$$\tilde{\text{ucb}}_{t-1}(\boldsymbol{x}) = \min_{1 \le i \le t} \text{ucb}_{i-1}(\boldsymbol{x}), \tilde{\text{lcb}}_{t-1}(\boldsymbol{x}) = \max_{1 \le i \le t} \text{lcb}_{i-1}(\boldsymbol{x}).$$

Conversely, we did not perform this operation in the infinite set setting, and calculated the AF instead using $\tilde{\text{ucb}}_{t-1}(\boldsymbol{x}) = \text{ucb}_{t-1}(\boldsymbol{x})$ and $\tilde{\text{lcb}}_{t-1}(\boldsymbol{x}) = \text{lcb}_{t-1}(\boldsymbol{x})$. Under this setup, one initial point was chosen at random and the algorithm was run for 500 iterations. This simulation was repeated 100 times and the average $r_t$ and Fscore$_t$ at each iteration were calculated.

From Fig 4, it can be confirmed that the proposed method has performance equal to or better than the comparison methods in terms of both $r_t$ and Fscore$_t$ in the sphere function setting. In the case of the Rosenbrock function setting, the proposed method exhibited performance equivalent to or better than the comparison method in terms of $r_t$. Moreover, in terms of Fscore$_t$, the Random method showed the best performance up to 250 iterations, but the proposed method matched or outperformed the comparison methods by the end of the iterations. In the Styblinski-Tang function setting, Random performed best in terms of $r_t$ and Fscore$_t$ up to around 300 iterations, but the proposed method equaled or surpassed the comparison methods by the final iterations.

### 5.3 Real-world Data Experiments

In this section, we conducted experiments using the carrier lifetime value, a measure of the quality performance of silicon ingots used in solar cells (Kutsukake et al., 2015). The data we used include the two-dimensional coordinates $\boldsymbol{x} = (x_1, x_2) \in \mathbb{R}^2$ of the sample surface and the carrier lifetime values $\tilde{f}(\boldsymbol{x}) \in [0.091587, 7.4613]$ at each coordinate, where $x_1 \in \{2a + 6 \mid 1 \le a \le 89\}$, $x_2 \in \{2a + 6 \mid 1 \le a \le 74\}$ and $|\mathcal{X}| = 89 \times 74 = 6586$. In quality evaluation, identifying defective regions, known as red zones areas where the value of $\tilde{f}(\boldsymbol{x})$ falls below a certain threshold is crucial. In this experiment, the threshold was set to 3, and we focused on identifying regions where $\tilde{f}(\boldsymbol{x})$ is 3 or less. We considered $f(\boldsymbol{x}) = -\tilde{f}(\boldsymbol{x}) + 3$ as the black-box function and performed experiments with $\theta = 0$. Additionally, the experiment was conducted assuming there was no noise in the observations. Moreover, to stabilize the posterior distribution calculation, $\sigma_{\text{noise}}^2 = 10^{-6}$ was used in the calculation. We used the following Matérn 3/2 kernel:

$$k(\boldsymbol{x}, \boldsymbol{x}') = 4\left(1 + \frac{\sqrt{3}\|\boldsymbol{x} - \boldsymbol{x}'\|_2}{25}\right)\exp\left(-\frac{\sqrt{3}\|\boldsymbol{x} - \boldsymbol{x}'\|_2}{25}\right).$$

The performance was evaluated using the loss $r_t$ and Fscore$_t$. As AFs, we compared six methods used in Section 5.1. We used $\beta_t^{1/2} = 3$ as the confidence parameter required for MILE and Straddle, and $\beta_t^{1/2} = \sqrt{2\log(6586 \times \pi^2 t^2/(6 \times 0.05))}$ for LSE. Under this setup, one initial point was chosen at random and the algorithm was run for 200 iterations. Because the observation noise was set to 0, the experiment was conducted under the setting that a point that had been observed once would not be observed thereafter. This simulation was repeated 100 times and the average $r_t$ and Fscore$_t$ at each iteration were calculated.

As shown in Fig. 5, the proposed method demonstrates performance that is equal to or better than the comparison methods in terms of both loss $r_t$ and Fscore$_t$.

## 6 Conclusion

In this study, we proposed a novel method called the randomized straddle algorithm, an extension of the straddle algorithm for LSE problems in black-box functions. The proposed method replaces the value of

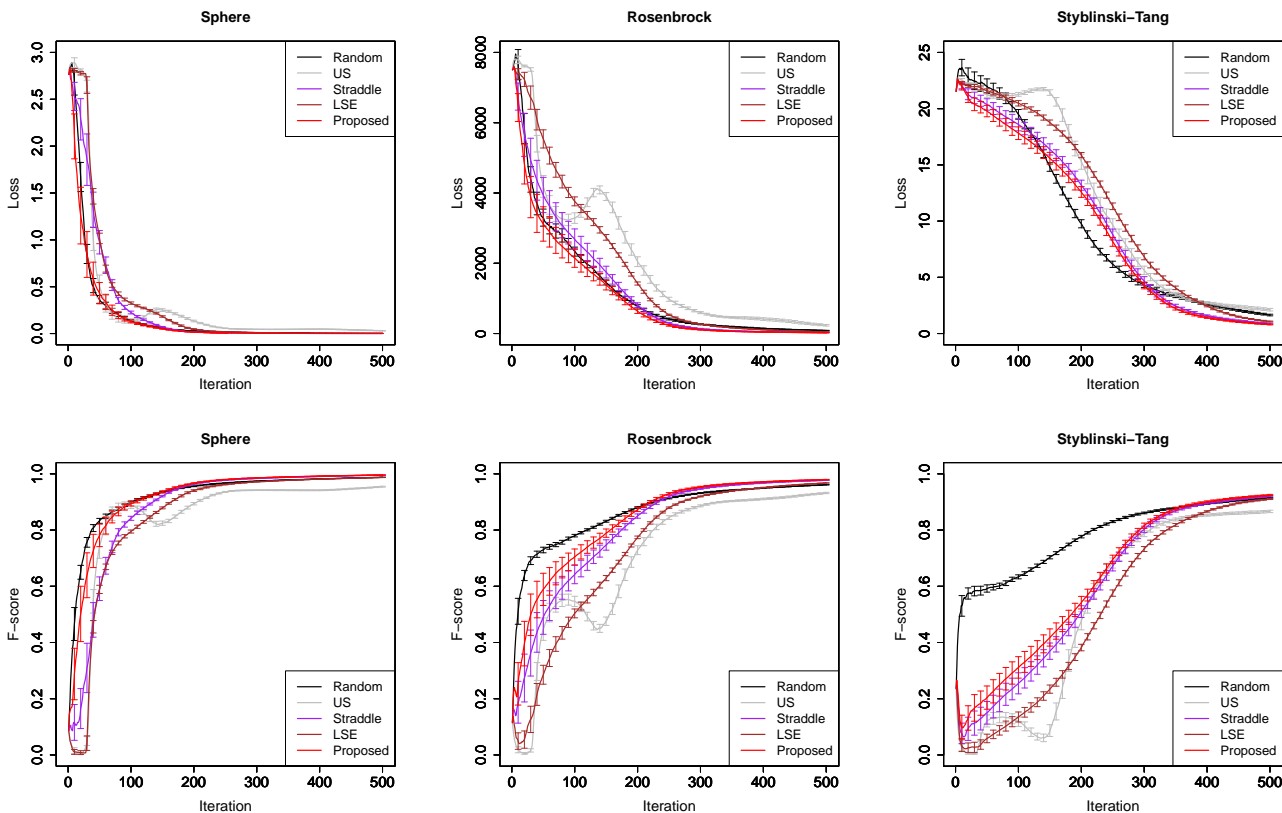

Figure 4: Averages of the loss $r_t$ and Fscore$_t$ for each AF over 100 simulations for each setting when the input space is infinite. The top row shows $r_t$, the bottom row shows Fscore$_t$, and each error bar length represents the six times the standard error.

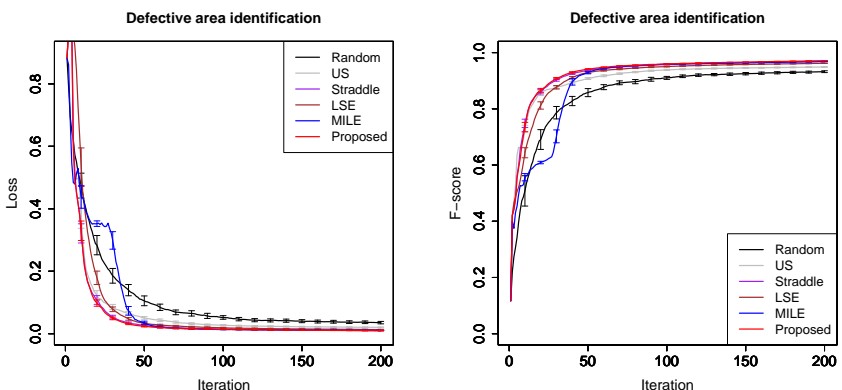

Figure 5: Averages of the loss $r_t$ and Fscore$_t$ for each AF over 100 simulations using the carrier lifetime data. The left figure shows $r_t$, while the right figure shows Fscore$_t$, with error bars representing six times the standard error.

$\beta_t$ in the straddle algorithm with a random sample from the chi-squared distribution with two degrees of freedom, performing LSE based on the GP posterior mean. As mensioned in Section 4, by considering an appropriate AF for an appropriate loss function and solving the difficult problem of appropriately designing

the distribution of the confidence parameters, we proved non-trivial theoretical results that the expected value of the loss in the estimated sets and that of the sum of losses are $O(\sqrt{\gamma_t/t})$ and $O(\sqrt{t\gamma_t})$, respectively.

Compared to existing methods, the proposed approach offers three key advantages. First, most theoretical analyses of existing methods involve confidence parameters that depend on the number of candidate points and iterations, whereas such terms are not present in the proposed method. Second, existing methods either do not apply to continuous search spaces or require discretization, with parameters for discretization often being unknown. In contrast, the proposed method is applicable to continuous search spaces without requiring algorithmic adjustments, providing the same theoretical guarantees as for finite search spaces. Third, while confidence parameters in existing methods tend to be overly conservative, the expected value of the confidence parameter in the proposed method is $\sqrt{2\pi}/2 \approx 1.25$, which is not excessively conservative. Furthermore, numerical experiments demonstrated that the performance of the proposed method is equal to or better than that of existing methods. This indicates that the proposed method performs comparably to heuristic methods while offering the added benefit of theoretical guarantees.

On the other hand, the proposed method has three drawbacks and limitations. First, since the proposed method does not make any significant changes other than adding randomization to the existing straddle, the practical performance is not dramatically improved compared to the case of using a fixed $\beta_t$. For example, if the next point is selected as the point at which $\mathbb{E}[r(H_t, L_t)]$ is the largest, practical performance can be expected to improve, but theoretical analysis in this case is not easy. Second, as mentioned in Section 4, the high-probability bounds derived by Theorem 4.3 are not tight. Finally, it is not easy to extend to other settings. As mentioned in Section 4, the theoretical analysis of the proposed method was achieved by appropriately selecting the loss, AF, and distribution of the confidence parameter. Therefore, simply replacing the parameters used in some AF with a chi-square distribution cannot directly apply the method to, for example, cases where the F-score is used as an evaluation index or to other problem settings. However, the insight gained from derivation of the theoretical results in this paper is that even if the problem setting is different and the evaluation index considered changes, if an AF is designed that can bound the evaluation index with high probability, it can be expected to be extended to other problem settings. In particular, both the results of Takeno et al. (2023) and the results of this paper use the property that some loss function can be bounded from above by $\beta_t^{1/2}\sigma_{t-1}(\boldsymbol{x}_t)$ with high probability. Therefore, it can be said that the key point of the theoretical analysis is to consider a combination of AF and loss that satisfies this property.

Future work includes resolving the above-mentioned drawbacks and limitations. In addition, since BO and LSE for black-box functions become difficult to handle when the input variables are high-dimensional, extending the proposed method to high-dimensional settings is also one of future work.

### Acknowledgments

This work was partially supported by JSPS KAKENHI (JP20H00601,JP23K16943,JP23K19967,JP24K20847), JST ACT-X (JPMJAX23CD), JST CREST (JPMJCR21D3, JPMJCR22N2), JST Moonshot R&D (JPMJMS2033-05), JST AIP Acceleration Research (JPMJCR21U2), NEDO (JPNP18002, JPNP20006) and RIKEN Center for Advanced Intelligence Project.

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

---

**Algorithm 2** Randomized Straddle Algorithms for Max-value Loss in the Finite Setting

---

**Input:** GP prior $\mathcal{GP}(0, k)$, threshold $\theta \in \mathbb{R}$
  **for** $t = 1, 2, \ldots, T$ **do**
    Compute $\mu_{t-1}(\boldsymbol{x})$ and $\sigma_{t-1}^2(\boldsymbol{x})$ for each $\boldsymbol{x} \in \mathcal{X}$ by equation 1
    Estimate $H_t$ and $L_t$ by equation 2
    Generate $\xi_t$ from the chi-squared distribution with two degrees of freedom
    Compute $\beta_t = \xi_t + 2\log(|\mathcal{X}|)$, $\mathrm{ucb}_{t-1}(\boldsymbol{x})$, $\mathrm{lcb}_{t-1}(\boldsymbol{x})$ and $\tilde{a}_{t-1}(\boldsymbol{x})$
    Select the next evaluation point $\boldsymbol{x}_t$ by $\boldsymbol{x}_t = \arg\max_{\boldsymbol{x} \in \mathcal{X}} \tilde{a}_{t-1}(\boldsymbol{x})$
    Observe $y_t = f(\boldsymbol{x}_t) + \varepsilon_t$ at the point $\boldsymbol{x}_t$
    Update GP by adding the observed data
  **end for**
**Output:** Return $H_{\hat{T}}$ and $L_{\hat{T}}$ as the estimated sets, where $\hat{T}$ is given by equation 6

---

# A Extension to Max-value Loss

In this section, we consider the following max-value loss defined based on the maximum value of $l_t(\boldsymbol{x})$:

$$r(H_t, L_t) = \max_{\boldsymbol{x} \in \mathcal{X}} l_t(\boldsymbol{x}) \equiv \tilde{r}_t.$$

When $\mathcal{X}$ is finite, we need to modify the definition of the AF and the estimated sets returned at the end of the algorithm. Conversely, if $\mathcal{X}$ is an infinite set, the definitions of $H_t$ and $L_t$ should be modified in addition to the above. Therefore, we discuss the finite and infinite cases separately.

## A.1 Proposed Method for Max-value Loss when $\mathcal{X}$ is Finite

When $\mathcal{X}$ is finite, we propose the following AF with a modified distribution that $\beta_t$ follows.

**Definition A.1** (Randomized Straddle for Max-value Loss)**.** For each $t \geq 1$, let $\xi_t$ be a random sample from the chi-squared distribution with two degrees of freedom, where $\xi_1, \ldots, \xi_t, \varepsilon_1, \ldots, \varepsilon_t, f$ are mutually independent. Define $\beta_t = \xi_t + 2\log(|\mathcal{X}|)$. Then, the randomized straddle AF for the max-value loss, $\tilde{a}_{t-1}(\boldsymbol{x})$, is defined as:

$$\tilde{a}_{t-1}(\boldsymbol{x}) = \max\{\min\{\mathrm{ucb}_{t-1}(\boldsymbol{x}) - \theta, \theta - \mathrm{lcb}_{t-1}(\boldsymbol{x})\}, 0\}. \tag{5}$$

By using $\tilde{a}_{t-1}(\boldsymbol{x})$, the next point to be evaluated is selected by $\boldsymbol{x}_t = \arg\max_{\boldsymbol{x} \in \mathcal{X}} \tilde{a}_{t-1}(\boldsymbol{x})$. Additionally, we change estimation sets returned at the end of iterations $T$ in the algorithm to the following instead of $H_T$ and $L_T$:

**Definition A.2.** For each $t$, define

$$\hat{t} = \arg\min_{1 \leq i \leq t} \mathbb{E}_t[\tilde{r}_i], \tag{6}$$

where $\mathbb{E}_t[\cdot]$ represents the conditional expectation given $D_{t-1}$. Then, at the end of iterations $T$, we define $H_{\hat{T}}$ and $L_{\hat{T}}$ to be the estimated sets.

Finally, we give the pseudocode of the proposed algorithm in Algorithm 2.

### A.1.1 Theoretical Analysis for Max-value Loss when $\mathcal{X}$ is Finite

For the max-value loss, the following theorem holds under Algorithm 2.

**Theorem A.1.** Let $f$ be a sample path from $\mathcal{GP}(0, k)$, where $k(\cdot, \cdot)$ is a positive-definite kernel satisfying $k(\boldsymbol{x}, \boldsymbol{x}) \leq 1$ for any $\boldsymbol{x} \in \mathcal{X}$. For each $t \geq 1$, let $\xi_t$ be a random sample from the chi-squared distribution with two degrees of freedom, where $\xi_1, \ldots, \xi_t, \varepsilon_1, \ldots, \varepsilon_t, f$ are mutually independent. Define $\beta_t = \xi_t + 2\log(|\mathcal{X}|)$. Then, the following holds for $\tilde{R}_t = \sum_{i=1}^{t} \tilde{r}_i$:

$$\mathbb{E}[\tilde{R}_t] \leq \sqrt{\tilde{C}_1 t \gamma_t},$$

where $\tilde{C}_1 = (4 + 4\log(|\mathcal{X}|))/\log(1 + \sigma_{\text{noise}}^{-2})$ and the expectation is taken with all randomness including $f, \varepsilon_t$ and $\beta_t$.

From Theorem A.1, the following theorem holds.

**Theorem A.2.** Under the assumptions of Theorem A.1, the following inequality holds:

$$\mathbb{E}[r_{\hat{t}}] \leq \sqrt{\frac{\tilde{C}_1 \gamma_t}{t}},$$

where $\hat{t}$ and $\tilde{C}_1$ are given in equation 6 and Theorem A.1, respectively.

Comparing Theorems 4.1 and A.1, when considering the max-value loss, $\beta_t$ should be $2\log(|\mathcal{X}|)$ larger than in the case of $r_t$, and the constant that appears in the upper bound of the expected value of the cumulative loss has the relationship $\tilde{C}_1 = (1 + \log(|\mathcal{X}|))C_1$. Note that while the upper bound for $r_t$ does not depend on $\mathcal{X}$, it depends on the logarithm of the number of elements in $\mathcal{X}$ for the max-value loss. Also, when comparing Theorem 4.2 and A.2, it is not necessary to consider $\hat{t}$ in $r_t$, whereas it is necessary to consider $\hat{t}$ in the max-value loss. For the max-value loss, it is difficult to analytically derive $\mathbb{E}_t[\tilde{r}_i]$, and hence, it is also difficult to precisely calculate $\hat{t}$. Nevertheless, because the posterior distribution of $f$ given $D_{t-1}$ is again a GP, we can generate $M$ sample paths from the GP posterior distribution and calculate the realization $\tilde{r}_i^{(j)}$ of $\tilde{r}_i$ from each sample path $f^{(j)}$, and calculate the estimate $\check{t}$ of $\hat{t}$ as

$$\check{t} = \arg\min_{1 \leq i \leq t} \frac{1}{M} \sum_{j=1}^{M} \tilde{r}_i^{(j)}.$$

### A.2 Proposed Method for Max-value Loss when $\mathcal{X}$ is Infinite

In this section, we assume that the input space $\mathcal{X} \subset \mathbb{R}^d$ is a compact set and satisfies $\mathcal{X} \subset [0, r]^d$, where $r > 0$. Furthermore, we assume the following additional assumption for $f$:

**Assumption A.1.** Let $f$ be differentiable with probability 1. Assuming positive constants $a, b$ exist, such that

$$\mathbb{P}\left(\sup_{\boldsymbol{x} \in \mathcal{X}} \left|\frac{\partial f}{\partial x_j}\right| > L\right) \leq a \exp\left(-\left(\frac{L}{b}\right)^2\right), \quad j \in [d],$$

where $x_j$ is the $j$-th element of $\boldsymbol{x}$ and $[d] \equiv \{1, \ldots, d\}$.

Next, we provide a LSE method based on the discretization of the input space.

#### A.2.1 Level Set Estimation for Max-value Loss when $\mathcal{X}$ is Infinite

For each $t \geq 1$, let $\mathcal{X}_t$ be a finite subset of $\mathcal{X}$. Also, for any $\boldsymbol{x} \in \mathcal{X}$, let $[\boldsymbol{x}]_t$ be the element of $\mathcal{X}_t$ that has the shortest $L_1$ distance from $\boldsymbol{x}$[4] Then, we define $H_t$ and $L_t$ as

$$H_t = \{\boldsymbol{x} \in \mathcal{X} \mid \mu_{t-1}([\boldsymbol{x}]_t) \geq \theta\}, \ L_t = \{\boldsymbol{x} \in \mathcal{X} \mid \mu_{t-1}([\boldsymbol{x}]_t) < \theta\}. \tag{7}$$

#### A.2.2 Acquisition Function for Max-value Loss when $\mathcal{X}$ is Infinite

We define a randomized straddle AF based on $\mathcal{X}_t$:

**Definition A.3.** For each $t \geq 1$, let $\xi_t$ be a random sample from the chi-squared distribution with two degrees of freedom, where $\xi_1, \ldots, \xi_t, \varepsilon_1, \ldots, \varepsilon_t, f$ are mutually independent. Define $\beta_t = 2\log(|\mathcal{X}_t|) + \xi_t$. Then, the randomized straddle AF for the max-value loss when $\mathcal{X}$ is infinite, $\check{a}_{t-1}(\boldsymbol{x})$, is defined as:

$$\check{a}_{t-1}(\boldsymbol{x}) = \max\{\min\{\text{ucb}_{t-1}(\boldsymbol{x}) - \theta, \theta - \text{lcb}_{t-1}(\boldsymbol{x})\}, 0\}.$$

The next point to be evaluated is selected by $\boldsymbol{x}_t = \arg\max_{\boldsymbol{x} \in \mathcal{X}} \check{a}_{t-1}(\boldsymbol{x})$. Finally, we give the pseudocode of the proposed algorithm in Algorithm 3.

---

[4]If there are multiple $\boldsymbol{x} \in \mathcal{X}_t$ with the shortest $L_1$ distance, determine the one that is unique. For example, we first choose the option with the smallest first component. If a unique determination is not possible, we then select the option with the smallest second component. This process is repeated up to the $d$-th component to achieve a unique determination.

---

**Algorithm 3** Randomized Straddle Algorithms for Max-value Loss in the Infinite Setting

---

**Input:** GP prior $\mathcal{GP}(0, k)$, threshold $\theta \in \mathbb{R}$, discretized sets $\mathcal{X}_1, \ldots, \mathcal{X}_T$
  **for** $t = 1, 2, \ldots, T$ **do**
    Compute $\mu_{t-1}(\boldsymbol{x})$ and $\sigma_{t-1}^2(\boldsymbol{x})$ for each $\boldsymbol{x} \in \mathcal{X}$ by equation 1
    Estimate $H_t$ and $L_t$ by equation 7
    Generate $\xi_t$ from the chi-squared distribution with two degrees of freedom
    Compute $\beta_t = \xi_t + 2\log(|\mathcal{X}_t|)$, $\mathrm{ucb}_{t-1}(\boldsymbol{x})$, $\mathrm{lcb}_{t-1}(\boldsymbol{x})$ and $\tilde{a}_{t-1}(\boldsymbol{x})$
    Select the next evaluation point $\boldsymbol{x}_t$ by $\boldsymbol{x}_t = \arg\max_{\boldsymbol{x} \in \mathcal{X}} \check{a}_{t-1}(\boldsymbol{x})$
    Observe $y_t = f(\boldsymbol{x}_t) + \varepsilon_t$ at the point $\boldsymbol{x}_t$
    Update GP by adding the observed data
  **end for**
**Output:** Return $H_{\hat{T}}$ and $L_{\hat{T}}$ as the estimated sets, where $\hat{T} = \arg\min_{1 \leq i \leq T} \mathbb{E}_T[\tilde{r}_i]$

---

### A.2.3 Theoretical Analysis for Max-value Loss when $\mathcal{X}$ is Infinite

Under Algorithm 3, the following theorem holds.

**Theorem A.3.** Let $\mathcal{X} \subset [0, r]^d$ be a compact set with $r > 0$. Assume that $f$ is a sample path from $\mathcal{GP}(0, k)$, where $k(\cdot, \cdot)$ is a positive-definite kernel satisfying $k(\boldsymbol{x}, \boldsymbol{x}) \leq 1$ for any $\boldsymbol{x} \in \mathcal{X}$. Also assume that Assumption A.1 holds. Moreover, for each $t \geq 1$, let $\tau_t = \lceil bdrt^2(\sqrt{\log(ad)} + \sqrt{\pi}/2) \rceil$, and let $\mathcal{X}_t$ be a finite subset of $\mathcal{X}$ satisfying $|\mathcal{X}_t| = \tau_t^d$ and

$$\|\boldsymbol{x} - [\boldsymbol{x}]_t\|_1 \leq \frac{dr}{\tau_t}, \quad \boldsymbol{x} \in \mathcal{X}.$$

Suppose that $\xi_t$ is a random sample from the chi-squared distribution with two degrees of freedom, where $\xi_1, \ldots, \xi_t, \varepsilon_1, \ldots, \varepsilon_t, f$ are mutually independent. Define $\beta_t = 2d\log(\lceil bdrt^2(\sqrt{\log(ad)} + \sqrt{\pi}/2) \rceil) + \xi_t$. Then, the following holds for $\tilde{R}_t = \sum_{i=1}^t \tilde{r}_i$:

$$\mathbb{E}[\tilde{R}_t] \leq \frac{\pi^2}{6} + \sqrt{\check{C}_1 t \gamma_t (2 + s_t)},$$

where $\check{C}_1 = 2/\log(1 + \sigma_{\text{noise}}^{-2})$ and $s_t = 2d\log(\lceil bdrt^2(\sqrt{\log(ad)} + \sqrt{\pi}/2) \rceil)$, and the expectation is taken with all randomness including $f, \varepsilon_t$ and $\beta_t$.

From Theorem A.3, the following holds.

**Theorem A.4.** Under the assumptions of Theorem A.3, define

$$\hat{t} = \arg\min_{1 \leq i \leq t} \mathbb{E}_t[\tilde{r}_i].$$

Then, the following holds:

$$\mathbb{E}[\tilde{r}_{\hat{t}}] \leq \frac{\pi^2}{6t} + \sqrt{\frac{\check{C}_1 \gamma_t (2 + s_t)}{t}},$$

where $\check{C}_1$ and $s_t$ are given in Theorem A.3.

## B Proofs

### B.1 Proof of Theorem 4.1

*Proof.* Let $\delta \in (0, 1)$. For any $t \geq 1$, $D_{t-1}$ and $\boldsymbol{x} \in \mathcal{X}$, from the proof of Lemma 5.1 in Srinivas et al. (2010), the following holds with probability at least $1 - \delta$:

$$\mathrm{lcb}_{t-1,\delta}(\boldsymbol{x}) \equiv \mu_{t-1}(\boldsymbol{x}) - \beta_\delta^{1/2}\sigma_{t-1}(\boldsymbol{x}) \leq f(\boldsymbol{x}) \leq \mu_{t-1}(\boldsymbol{x}) + \beta_\delta^{1/2}\sigma_{t-1}(\boldsymbol{x}) \equiv \mathrm{ucb}_{t-1,\delta}(\boldsymbol{x}), \tag{8}$$

where $\beta_\delta = 2\log(1/\delta)$. Here, we consider the case where $\boldsymbol{x} \in H_t$. If $\boldsymbol{x} \in H^*$, we have $l_t(\boldsymbol{x}) = 0$. In contrast, if $\boldsymbol{x} \in L^*$, noting that $\mathrm{lcb}_{t-1,\delta}(\boldsymbol{x}) \leq f(\boldsymbol{x})$ by equation 8 we get

$$l_t(\boldsymbol{x}) = \theta - f(\boldsymbol{x}) \leq \theta - \mathrm{lcb}_{t-1,\delta}(\boldsymbol{x}).$$

Moreover, the inequality $\mu_{t-1}(\boldsymbol{x}) \geq \theta$ holds because $\boldsymbol{x} \in H_t$. Hence, from the definition of $\mathrm{lcb}_{t-1,\delta}(\boldsymbol{x})$ and $\mathrm{ucb}_{t-1,\delta}(\boldsymbol{x})$, we obtain

$$\theta - \mathrm{lcb}_{t-1,\delta}(\boldsymbol{x}) \leq \mathrm{ucb}_{t-1,\delta}(\boldsymbol{x}) - \theta.$$

Therefore, we get

$$\begin{aligned} l_t(\boldsymbol{x}) \leq \theta - \mathrm{lcb}_{t-1,\delta}(\boldsymbol{x}) &= \min\{\mathrm{ucb}_{t-1,\delta}(\boldsymbol{x}) - \theta, \theta - \mathrm{lcb}_{t-1,\delta}(\boldsymbol{x})\} \\ &\leq \max\{\min\{\mathrm{ucb}_{t-1,\delta}(\boldsymbol{x}) - \theta, \theta - \mathrm{lcb}_{t-1,\delta}(\boldsymbol{x})\}, 0\} \equiv a_{t-1,\delta}(\boldsymbol{x}). \end{aligned}$$

Similarly, we consider the case where $\boldsymbol{x} \in L_t$. If $\boldsymbol{x} \in L^*$, we obtain $l_t(\boldsymbol{x}) = 0$. Thus, because $a_{t-1,\delta}(\boldsymbol{x}) \geq 0$, we get $l_t(\boldsymbol{x}) \leq a_{t-1,\delta}(\boldsymbol{x})$. Moreover, if $\boldsymbol{x} \in H^*$, noting that $f(\boldsymbol{x}) \leq \mathrm{ucb}_{t-1,\delta}(\boldsymbol{x})$ by equation 8, we obtain

$$l_t(\boldsymbol{x}) = f(\boldsymbol{x}) - \theta \leq \mathrm{ucb}_{t-1,\delta}(\boldsymbol{x}) - \theta.$$

Here, the inequality $\mu_{t-1}(\boldsymbol{x}) < \theta$ holds because $\boldsymbol{x} \in L_t$. Therefore, from the definition of $\mathrm{lcb}_{t-1,\delta}(\boldsymbol{x})$ and $\mathrm{ucb}_{t-1,\delta}(\boldsymbol{x})$, we obtain

$$\mathrm{ucb}_{t-1,\delta}(\boldsymbol{x}) - \theta \leq \theta - \mathrm{lcb}_{t-1,\delta}(\boldsymbol{x}).$$

Thus, the following inequality holds:

$$l_t(\boldsymbol{x}) \leq \mathrm{ucb}_{t-1,\delta}(\boldsymbol{x}) - \theta = \min\{\mathrm{ucb}_{t-1,\delta}(\boldsymbol{x}) - \theta, \theta - \mathrm{lcb}_{t-1,\delta}(\boldsymbol{x})\} \leq a_{t-1,\delta}(\boldsymbol{x}).$$

Therefore, for all cases, the inequality $l_t(\boldsymbol{x}) \leq a_{t-1,\delta}(\boldsymbol{x})$ holds. This indicates that the following inequality holds with probability at least $1 - \delta$:

$$l_t(\boldsymbol{x}) \leq a_{t-1,\delta}(\boldsymbol{x}) \leq \max_{\tilde{\boldsymbol{x}} \in \mathcal{X}} a_{t-1,\delta}(\tilde{\boldsymbol{x}}). \tag{9}$$

Next, we consider the conditional distribution of $l_t(\boldsymbol{x})$ given $D_{t-1}$. Note that this distribution does not depend on $\beta_\delta$. Let $F_{t-1}(\cdot)$ be a distribution function of $l_t(\boldsymbol{x})$ given $D_{t-1}$. Then, from equation 9 we have

$$F_{t-1}\left(\max_{\tilde{\boldsymbol{x}} \in \mathcal{X}} a_{t-1,\delta}(\tilde{\boldsymbol{x}})\right) \geq 1 - \delta.$$

Hence, by considering the generalized inverse function of $F_{t-1}(\cdot)$ for both sides, the following inequality holds:

$$F_{t-1}^{-1}(1 - \delta) \leq \max_{\tilde{\boldsymbol{x}} \in \mathcal{X}} a_{t-1,\delta}(\tilde{\boldsymbol{x}}).$$

Here, if $\delta$ follows the uniform distribution on the interval $(0, 1)$, then $1 - \delta$ follows the same distribution. In this case, the distribution of $F_{t-1}^{-1}(1 - \delta)$ is equal to the distribution of $l_t(\boldsymbol{x})$ given $D_{t-1}$. This implies that

$$\mathbb{E}_t[l_t(\boldsymbol{x})] \leq \mathbb{E}_\delta\left[\max_{\boldsymbol{x} \in \mathcal{X}} a_{t-1,\delta}(\boldsymbol{x})\right],$$

where $\mathbb{E}_\delta[\cdot]$ means the expectation with respect to $\delta$. Furthermore, because $2\log(1/\delta)$ and $\beta_t$ follow the chi-squared distribution with two degrees of freedom, the following holds:

$$\mathbb{E}_t[l_t(\boldsymbol{x})] \leq \mathbb{E}_{\beta_t}[a_{t-1}(\boldsymbol{x}_t)].$$

Thus, if $\mathcal{X}$ is finite, from the definition of $r_t$ we obtain

$$\mathbb{E}_t[r_t] = \mathbb{E}_t\left[\frac{1}{|\mathcal{X}|}\sum_{\boldsymbol{x} \in \mathcal{X}} l_t(\boldsymbol{x})\right] = \frac{1}{|\mathcal{X}|}\sum_{\boldsymbol{x} \in \mathcal{X}}\mathbb{E}_t[l_t(\boldsymbol{x})] \leq \frac{1}{|\mathcal{X}|}\sum_{\boldsymbol{x} \in \mathcal{X}}\mathbb{E}_{\beta_t}[a_{t-1}(\boldsymbol{x}_t)] = \mathbb{E}_{\beta_t}[a_{t-1}(\boldsymbol{x}_t)].$$

Similarly, if $\mathcal{X}$ is infinite, from the definition of $r_t$ and non-negativity of $l_t(\boldsymbol{x})$, using Fubini's theorem we get

$$\mathbb{E}_t[r_t] = \mathbb{E}_t\left[\frac{1}{\mathrm{Vol}(\mathcal{X})}\int_{\mathcal{X}} l_t(\boldsymbol{x})\mathrm{d}\boldsymbol{x}\right] = \frac{1}{\mathrm{Vol}(\mathcal{X})}\int_{\mathcal{X}}\mathbb{E}_t[l_t(\boldsymbol{x})]\mathrm{d}\boldsymbol{x} \le \frac{1}{\mathrm{Vol}(\mathcal{X})}\int_{\mathcal{X}}\mathbb{E}_{\beta_t}[a_{t-1}(\boldsymbol{x}_t)]\,\mathrm{d}\boldsymbol{x} = \mathbb{E}_{\beta_t}[a_{t-1}(\boldsymbol{x}_t)].$$

Therefore, the inequality $\mathbb{E}_t[r_t] \le \mathbb{E}_{\beta_t}[a_{t-1}(\boldsymbol{x}_t)]$ holds for both cases. Moreover, from the definition of $a_{t-1}(\boldsymbol{x})$, the following inequality holds:

$$a_{t-1}(\boldsymbol{x}_t) \le \beta_t^{1/2}\sigma_{t-1}(\boldsymbol{x}_t)$$

Hence, we get the following inequality:

$$\mathbb{E}[R_t] = \mathbb{E}\left[\sum_{i=1}^t r_i\right] \le \mathbb{E}\left[\sum_{i=1}^t \beta_i^{1/2}\sigma_{i-1}(\boldsymbol{x}_i)\right]$$

$$\xrightarrow{\text{Cauchy-Schwarz inequality}} \le \mathbb{E}\left[\left(\sum_{i=1}^t \beta_i\right)^{1/2}\left(\sum_{i=1}^t \sigma_{i-1}^2(\boldsymbol{x}_i)\right)^{1/2}\right]$$

$$\xrightarrow{\text{Hölder's inequality}} \le \sqrt{\mathbb{E}\left[\sum_{i=1}^t \beta_i\right]}\sqrt{\mathbb{E}\left[\sum_{i=1}^t \sigma_{i-1}^2(\boldsymbol{x}_i)\right]}$$

$$\xrightarrow{\mathbb{E}[\beta_i]=2} = \sqrt{2t}\sqrt{\mathbb{E}\left[\sum_{i=1}^t \sigma_{i-1}^2(\boldsymbol{x}_i)\right]}$$

$$\le \sqrt{2t}\sqrt{\mathbb{E}\left[\frac{2}{\log(1+\sigma_{\mathrm{noise}}^{-2})}\gamma_t\right]}$$

$$= \sqrt{C_1 t \gamma_t},$$

where the last inequality is derived by the proof of Lemma 5.4 in Srinivas et al. (2010). □

## B.2 Proof of Theorem 4.2

We first give three lemmas to prove Theorem 4.2. Theorem 4.2 is proved by Lemma B.1 and B.3.

**Lemma B.1.** Under the assumptions of Theorem 4.1, let

$$\hat{t} = \operatorname*{arg\,min}_{1\le i\le t} \mathbb{E}_t[r_i].$$

Then, the following inequality holds:

$$\mathbb{E}[r_{\hat{t}}] \le \sqrt{\frac{C_1\gamma_t}{t}}.$$

*Proof.* From the definition of $\hat{t}$, the inequality $\mathbb{E}_t[r_{\hat{t}}] \le \frac{\sum_{i=1}^t \mathbb{E}_t[r_i]}{t}$ holds. Therefore, we obtain

$$\mathbb{E}[r_{\hat{t}}] \le \frac{\sum_{i=1}^t \mathbb{E}[r_i]}{t} = \frac{\mathbb{E}\left[\sum_{i=1}^t r_i\right]}{t} = \frac{\mathbb{E}[R_t]}{t}.$$

By combining this and Theorem 4.1, we get the desired result. □

**Lemma B.2.** For any $t \ge 1$, $i \le t$ and $\boldsymbol{x} \in \mathcal{X}$, the expectation $\mathbb{E}_t[l_i(\boldsymbol{x})]$ can be calculated as follows:

$$\mathbb{E}_t[l_i(\boldsymbol{x})] = \begin{cases} \sigma_{t-1}(\boldsymbol{x})\left[\phi(-\alpha) + \alpha\left\{1 - \Phi(-\alpha)\right\}\right] & \text{if } \boldsymbol{x} \in L_i \\ \sigma_{t-1}(\boldsymbol{x})\left[\phi(\alpha) - \alpha\left\{1 - \Phi(\alpha)\right\}\right] & \text{if } \boldsymbol{x} \in H_i \end{cases},$$

where $\alpha = \frac{\mu_{t-1}(\boldsymbol{x})-\theta}{\sigma_{t-1}(\boldsymbol{x})}$, and $\phi(z)$ and $\Phi(z)$ are the density and distribution function of the standard normal distribution, respectively.

*Proof.* From the definition of $l_i(\boldsymbol{x})$, if $\boldsymbol{x} \in L_i$, $l_i(\boldsymbol{x})$ can be expressed as $l_i(\boldsymbol{x}) = (f(\boldsymbol{x}) - \theta)\mathbb{1}[f(\boldsymbol{x}) \geq \theta]$, where $\mathbb{1}[\cdot]$ is the indicator function which takes 1 if the condition $\cdot$ holds, otherwise 0. Furthermore, the conditional distribution of $f(\boldsymbol{x})$ given $D_{t-1}$ is the normal distribution with mean $\mu_{t-1}(\boldsymbol{x})$ and variance $\sigma_{t-1}^2(\boldsymbol{x})$. Thus, from the definition of $\mathbb{E}_t[\cdot]$, the following holds:

$$
\begin{aligned}
\mathbb{E}_t[l_i(\boldsymbol{x})] &= \int_\theta^\infty (y - \theta) \frac{1}{\sqrt{2\pi\sigma_{t-1}^2(\boldsymbol{x})}} \exp\left(-\frac{(y - \mu_{t-1}(\boldsymbol{x}))^2}{2\sigma_{t-1}^2(\boldsymbol{x})}\right) \mathrm{d}y \\
&= \int_\theta^\infty \sigma_{t-1}(\boldsymbol{x}) \left(\frac{y - \mu_{t-1}(\boldsymbol{x})}{\sigma_{t-1}(\boldsymbol{x})} + \frac{\mu_{t-1}(\boldsymbol{x}) - \theta}{\sigma_{t-1}(\boldsymbol{x})}\right) \frac{1}{\sqrt{2\pi\sigma_{t-1}^2(\boldsymbol{x})}} \exp\left(-\frac{(y - \mu_{t-1}(\boldsymbol{x}))^2}{2\sigma_{t-1}^2(\boldsymbol{x})}\right) \mathrm{d}y \\
&= \int_{-\alpha}^\infty \sigma_{t-1}(\boldsymbol{x}) (z + \alpha) \frac{1}{\sqrt{2\pi}} \exp\left(-\frac{z^2}{2}\right) \mathrm{d}z \\
&= \sigma_{t-1}(\boldsymbol{x}) \int_{-\alpha}^\infty (z + \alpha) \phi(z) \mathrm{d}z = \sigma_{t-1}(\boldsymbol{x})\{[-\phi(z)]_{-\alpha}^\infty + \alpha(1 - \Phi(-\alpha))\} \\
&= \sigma_{t-1}(\boldsymbol{x}) \left[\phi(-\alpha) + \alpha\{1 - \Phi(-\alpha)\}\right].
\end{aligned}
$$

Similarly, if $\boldsymbol{x} \in H_i$, $l_i(\boldsymbol{x})$ can be expressed as $l_i(\boldsymbol{x}) = (\theta - f(\boldsymbol{x}))\mathbb{1}[f(\boldsymbol{x}) < \theta]$. Then, we obtain

$$
\begin{aligned}
\mathbb{E}_t[l_i(\boldsymbol{x})] &= \int_{-\infty}^\theta (\theta - y) \frac{1}{\sqrt{2\pi\sigma_{t-1}^2(\boldsymbol{x})}} \exp\left(-\frac{(y - \mu_{t-1}(\boldsymbol{x}))^2}{2\sigma_{t-1}^2(\boldsymbol{x})}\right) \mathrm{d}y \\
&= \int_{-\infty}^\theta \sigma_{t-1}(\boldsymbol{x}) \left(\frac{\theta - \mu_{t-1}(\boldsymbol{x})}{\sigma_{t-1}(\boldsymbol{x})} + \frac{\mu_{t-1}(\boldsymbol{x}) - y}{\sigma_{t-1}(\boldsymbol{x})}\right) \frac{1}{\sqrt{2\pi\sigma_{t-1}^2(\boldsymbol{x})}} \exp\left(-\frac{(y - \mu_{t-1}(\boldsymbol{x}))^2}{2\sigma_{t-1}^2(\boldsymbol{x})}\right) \mathrm{d}y \\
&= \int_\infty^\alpha \sigma_{t-1}(\boldsymbol{x}) (z - \alpha) \frac{1}{\sqrt{2\pi}} \exp\left(-\frac{z^2}{2}\right) (-1)\mathrm{d}z \\
&= \sigma_{t-1}(\boldsymbol{x}) \int_\alpha^\infty (z - \alpha) \phi(z) \mathrm{d}z = \sigma_{t-1}(\boldsymbol{x})\{[-\phi(z)]_\alpha^\infty - \alpha(1 - \Phi(\alpha))\} \\
&= \sigma_{t-1}(\boldsymbol{x}) \left[\phi(\alpha) - \alpha\{1 - \Phi(\alpha)\}\right].
\end{aligned}
$$

$\square$

**Lemma B.3.** Under the assumptions of Theorem 4.1 the equality $\hat{t} = t$ holds.

*Proof.* Let $\boldsymbol{x} \in \mathcal{X}$. If $\boldsymbol{x} \in H_t$, the inequality $\mu_{t-1}(\boldsymbol{x}) \geq \theta$ holds. This implies that $\alpha \geq 0$. Hence, from Lemma B.2 we obtain

$$
\mathbb{E}_t[l_t(\boldsymbol{x})] = \sigma_{t-1}(\boldsymbol{x}) \left[\phi(\alpha) - \alpha\{1 - \Phi(\alpha)\}\right].
$$

Thus, since $\alpha \geq 0$, the following inequality holds:

$$
\sigma_{t-1}(\boldsymbol{x}) \left[\phi(\alpha) - \alpha\{1 - \Phi(\alpha)\}\right] \leq \sigma_{t-1}(\boldsymbol{x}) \left[\phi(-\alpha) + \alpha\{1 - \Phi(-\alpha)\}\right].
$$

Therefore, from the definition of $\mathbb{E}_t[l_i(\boldsymbol{x})]$, we get

$$
\mathbb{E}_t[l_t(\boldsymbol{x})] = \sigma_{t-1}(\boldsymbol{x}) \left[\phi(\alpha) - \alpha\{1 - \Phi(\alpha)\}\right] \leq \mathbb{E}_t[l_i(\boldsymbol{x})].
$$

Similarly, if $\boldsymbol{x} \in L_t$, using the same argument we have

$$
\mathbb{E}_t[l_t(\boldsymbol{x})] = \sigma_{t-1}(\boldsymbol{x}) \left[\phi(-\alpha) + \alpha\{1 - \Phi(-\alpha)\}\right] \leq \mathbb{E}_t[l_i(\boldsymbol{x})].
$$

Here, if $\mathcal{X}$ is finite, from the definition of $r_i$ we obtain

$$
\mathbb{E}_t[r_t] = \mathbb{E}_t\left[\frac{1}{|\mathcal{X}|}\sum_{\boldsymbol{x}\in\mathcal{X}} l_t(\boldsymbol{x})\right] = \frac{1}{|\mathcal{X}|}\sum_{\boldsymbol{x}\in\mathcal{X}} \mathbb{E}_t[l_t(\boldsymbol{x})] \leq \frac{1}{|\mathcal{X}|}\sum_{\boldsymbol{x}\in\mathcal{X}} \mathbb{E}_t[l_i(\boldsymbol{x})] = \mathbb{E}_t[r_i].
$$

Similarly, if $\mathcal{X}$ is infinite, by using the same argument and Fubini's theorem, we get $\mathbb{E}_t[r_t] \leq \mathbb{E}_t[r_i]$. Therefore, for all cases the inequality $\mathbb{E}_t[r_t] \leq \mathbb{E}_t[r_i]$ holds. This implies that $\hat{t} = t$. $\square$

From Lemma B.1 and B.3, we get Theorem 4.2.

### B.3 Proof of Theorem A.1

*Proof.* Let $\delta \in (0, 1)$. For any $t \geq 1$ and $D_{t-1}$, from the proof of Lemma 5.1 in Srinivas et al. (2010), with probability at least $1 - \delta$, the following holds for any $\boldsymbol{x} \in \mathcal{X}$:

$$\text{lcb}_{t-1,\delta}(\boldsymbol{x}) \equiv \mu_{t-1}(\boldsymbol{x}) - \beta_\delta^{1/2}\sigma_{t-1}(\boldsymbol{x}) \leq f(\boldsymbol{x}) \leq \mu_{t-1}(\boldsymbol{x}) + \beta_\delta^{1/2}\sigma_{t-1}(\boldsymbol{x}) \equiv \text{ucb}_{t-1,\delta}(\boldsymbol{x}),$$

where $\beta_\delta = 2\log(|\mathcal{X}|/\delta)$. Here, by using the same argument as in the proof of Theorem 4.1, the inequality $l_t(\boldsymbol{x}) \leq \tilde{a}_{t-1,\delta}(\boldsymbol{x})$ holds. Hence, the following holds with probability at least $1 - \delta$:

$$\tilde{r}_t = \max_{\boldsymbol{x}\in\mathcal{X}} l_t(\boldsymbol{x}) \leq \max_{\boldsymbol{x}\in\mathcal{X}} \tilde{a}_{t-1,\delta}(\boldsymbol{x}). \tag{10}$$

Next, we consider the conditional distribution of $\tilde{r}_t$ given $D_{t-1}$. Note that this distribution does not depend on $\beta_\delta$. Let $F_{t-1}(\cdot)$ be a distribution function of $\tilde{r}_t$ given $D_{t-1}$. Then, from equation 10, we obtain

$$F_{t-1}\left(\max_{\boldsymbol{x}\in\mathcal{X}} \tilde{a}_{t-1,\delta}(\boldsymbol{x})\right) \geq 1 - \delta.$$

Therefore, by taking the generalized inverse function for both sides, we get

$$F_{t-1}^{-1}(1-\delta) \leq \max_{\boldsymbol{x}\in\mathcal{X}} \tilde{a}_{t-1,\delta}(\boldsymbol{x}).$$

Here, if $\delta$ follows the uniform distribution on the interval $(0,1)$, $1-\delta$ follows the same distribution. Furthermore, since the distribution of $F_{t-1}^{-1}(1-\delta)$ is equal to the conditional distribution of $\tilde{r}_t$ given $D_{t-1}$, we have

$$\mathbb{E}_t[\tilde{r}_t] \leq \mathbb{E}_\delta\left[\max_{\boldsymbol{x}\in\mathcal{X}} \tilde{a}_{t-1,\delta}(\boldsymbol{x})\right].$$

Moreover, noting that $2\log(|\mathcal{X}|/\delta)$ and $\beta_t$ follow the same distribution, we obtain

$$\mathbb{E}_t[\tilde{r}_t] \leq \mathbb{E}_{\beta_t}[\tilde{a}_{t-1}(\boldsymbol{x}_t)].$$

Additionally, from $\tilde{a}_{t-1}(\boldsymbol{x})$, the following inequality holds:

$$\tilde{a}_{t-1}(\boldsymbol{x}_t) \leq \beta_t^{1/2}\sigma_{t-1}(\boldsymbol{x}_t).$$

Therefore, since $\mathbb{E}[\beta_t] = 2 + 2\log(|\mathcal{X}|))$, the following inequality holds:

$$\begin{aligned}
\mathbb{E}[\tilde{R}_t] = \mathbb{E}\left[\sum_{i=1}^t \tilde{r}_i\right] &\leq \mathbb{E}\left[\sum_{i=1}^t \beta_i^{1/2}\sigma_{i-1}(\boldsymbol{x}_i)\right] \\
&\leq \mathbb{E}\left[\left(\sum_{i=1}^t \beta_i\right)^{1/2}\left(\sum_{i=1}^t \sigma_{i-1}^2(\boldsymbol{x}_i)\right)^{1/2}\right] \\
&\leq \sqrt{\mathbb{E}\left[\sum_{i=1}^t \beta_i\right]}\sqrt{\mathbb{E}\left[\sum_{i=1}^t \sigma_{i-1}^2(\boldsymbol{x}_i)\right]} \\
&\leq \sqrt{t(2 + 2\log(|\mathcal{X}|))}\sqrt{\mathbb{E}\left[\sum_{i=1}^t \sigma_{i-1}^2(\boldsymbol{x}_i)\right]} \\
&\leq \sqrt{t(2 + 2\log(|\mathcal{X}|))}\sqrt{\mathbb{E}\left[\frac{2}{\log(1 + \sigma_{\text{noise}}^{-2})}\gamma_t\right]} \\
&= \sqrt{\tilde{C}_1 t \gamma_t}.
\end{aligned}$$

$\square$

### B.4 Proof of Theorem A.2

*Proof.* Theorem A.2 is proved by using the same argument as in the proof of Lemma B.1. □

### B.5 Proof of Theorem A.3

*Proof.* Let $\boldsymbol{x} \in \mathcal{X}$. If $\boldsymbol{x} \in H^* \cap H_t$ or $\boldsymbol{x} \in L^* \cap L_t$, the equality $l_t(\boldsymbol{x}) = 0$ holds. Hence, the following inequality holds:

$$l_t(\boldsymbol{x}) \leq l_t([\boldsymbol{x}]_t) \leq l_t([\boldsymbol{x}]_t) + |f(\boldsymbol{x}) - f([\boldsymbol{x}]_t)|.$$

We consider the case where $\boldsymbol{x} \in H^*$ and $\boldsymbol{x} \in L_t$, that is, $l_t(\boldsymbol{x}) = f(\boldsymbol{x}) - \theta$. Here, since $\boldsymbol{x} \in L_t$, the inequality $\mu_{t-1}([\boldsymbol{x}]_t) < \theta$ holds. This implies that $[\boldsymbol{x}]_t \in L_t$. If $[\boldsymbol{x}]_t \in H^*$, noting that $l_t([\boldsymbol{x}]_t) = f([\boldsymbol{x}]_t) - \theta$ we get

$$l_t(\boldsymbol{x}) = f(\boldsymbol{x}) - \theta = f(\boldsymbol{x}) - f([\boldsymbol{x}]_t) + f([\boldsymbol{x}]_t) - \theta \leq f([\boldsymbol{x}]_t) - \theta + |f(\boldsymbol{x}) - f([\boldsymbol{x}]_t)| = l_t([\boldsymbol{x}]_t) + |f(\boldsymbol{x}) - f([\boldsymbol{x}]_t)|.$$

Similarly, if $[\boldsymbol{x}]_t \in L^*$, noting that $f([\boldsymbol{x}]_t) < \theta$ and $0 \leq l_t([\boldsymbol{x}]_t)$ we obtain

$$l_t(\boldsymbol{x}) = f(\boldsymbol{x}) - \theta = f([\boldsymbol{x}]_t) - \theta + f(\boldsymbol{x}) - f([\boldsymbol{x}]_t) \leq 0 + f(\boldsymbol{x}) - f([\boldsymbol{x}]_t) \leq l_t([\boldsymbol{x}]_t) + |f(\boldsymbol{x}) - f([\boldsymbol{x}]_t)|.$$

Next, we consider the case where $\boldsymbol{x} \in L^*$ and $\boldsymbol{x} \in H_t$, that is, $l_t(\boldsymbol{x}) = \theta - f(\boldsymbol{x})$. Here, since $\boldsymbol{x} \in H_t$, the inequality $\mu_{t-1}([\boldsymbol{x}]_t) \geq \theta$holds. This implies that $[\boldsymbol{x}]_t \in H_t$. If $[\boldsymbol{x}]_t \in L^*$, noting that $l_t([\boldsymbol{x}]_t) = \theta - f([\boldsymbol{x}]_t)$, we have

$$l_t(\boldsymbol{x}) = \theta - f(\boldsymbol{x}) = \theta - f([\boldsymbol{x}]_t) + f([\boldsymbol{x}]_t) - f(\boldsymbol{x}) \leq l_t([\boldsymbol{x}]_t) + |f(\boldsymbol{x}) - f([\boldsymbol{x}]_t)|$$

Similarly, if $[\boldsymbol{x}]_t \in H^*$, noting that $f([\boldsymbol{x}]_t) \geq \theta$ and $0 \leq l_t([\boldsymbol{x}]_t)$, we get

$$l_t(\boldsymbol{x}) = \theta - f(\boldsymbol{x}) = \theta - f([\boldsymbol{x}]_t) + f([\boldsymbol{x}]_t) - f(\boldsymbol{x}) \leq 0 + f([\boldsymbol{x}]_t) - f(\boldsymbol{x}) \leq l_t([\boldsymbol{x}]_t) + |f(\boldsymbol{x}) - f([\boldsymbol{x}]_t)|.$$

Therefore, for all cases the following inequality holds:

$$l_t(\boldsymbol{x}) \leq l_t([\boldsymbol{x}]_t) + |f(\boldsymbol{x}) - f([\boldsymbol{x}]_t)|.$$

Here, let $L_{\max} = \sup_{j \in [d]} \sup_{\boldsymbol{x} \in \mathcal{X}} \left| \frac{\partial f}{\partial x_j} \right|$. Then, the following holds:

$$|f(\boldsymbol{x}) - f([\boldsymbol{x}]_t)| \leq L_{\max} \|\boldsymbol{x} - [\boldsymbol{x}]_t\|_1 \leq L_{\max} \frac{dr}{\tau_t}.$$

Thus, noting that

$$l_t(\boldsymbol{x}) \leq l_t([\boldsymbol{x}]_t) + L_{\max} \frac{dr}{\tau_t}$$

we obtain

$$\tilde{r}_t = \max_{\boldsymbol{x} \in \mathcal{X}} l_t(\boldsymbol{x}) \leq L_{\max} \frac{dr}{\tau_t} + \max_{\boldsymbol{x} \in \mathcal{X}} l_t([\boldsymbol{x}]_t) \equiv L_{\max} \frac{dr}{\tau_t} + \max_{\tilde{\boldsymbol{x}} \in \mathcal{X}_t} l_t(\tilde{\boldsymbol{x}}) \equiv L_{\max} \frac{dr}{\tau_t} + \check{r}_t.$$

In addition, from Lemma H.1 in Takeno et al. (2023), the following inequality holds:

$$\mathbb{E}[L_{\max}] \leq b(\sqrt{\log(ad)} + \sqrt{\pi}/2).$$

Hence, we get

$$\mathbb{E}\left[ L_{\max} \frac{dr}{\tau_t} \right] \leq \frac{b(\sqrt{\log(ad)} + \sqrt{\pi}/2)}{\tau_t} dr = \frac{b(\sqrt{\log(ad)} + \sqrt{\pi}/2)}{\lceil bdrt^2(\sqrt{\log(ad)} + \sqrt{\pi}/2) \rceil} dr$$

$$\leq \frac{b(\sqrt{\log(ad)} + \sqrt{\pi}/2)}{bdrt^2(\sqrt{\log(ad)} + \sqrt{\pi}/2)} dr = \frac{1}{t^2}.$$

Therefore, the following inequality holds:

$$\mathbb{E}[\tilde{R}_t] = \mathbb{E}\left[\sum_{i=1}^{t} \tilde{r}_i\right] \leq \sum_{i=1}^{t} \frac{1}{i^2} + \mathbb{E}\left[\sum_{i=1}^{t} \check{r}_i\right] \leq \frac{\pi^2}{6} + \mathbb{E}\left[\sum_{i=1}^{t} \check{r}_i\right].$$

Here, $\check{r}_i$ is the maximum value of the loss $l_i(\tilde{\boldsymbol{x}})$ restricted on $\mathcal{X}_i$, and since $\mathcal{X}_i$ is a finite set, by replacing $\mathcal{X}$ with $\mathcal{X}_i$ in the proof of Theorem A.1 and performing the same proof, we obtain $\mathbb{E}_i[\check{r}_i] \leq \mathbb{E}_\delta[\max_{\tilde{\boldsymbol{x}} \in \mathcal{X}_t} \check{a}_{i-1}(\tilde{\boldsymbol{x}})]$. Furthermore, since the next point to be evaluated is selected from $\mathcal{X}$, the following inequality holds:

$$\mathbb{E}_i[\check{r}_i] \leq \mathbb{E}_\delta[\max_{\tilde{\boldsymbol{x}} \in \mathcal{X}_t} \check{a}_{i-1}(\tilde{\boldsymbol{x}})] \leq \mathbb{E}_\delta[\max_{\boldsymbol{x} \in \mathcal{X}} \check{a}_{i-1}(\boldsymbol{x})].$$

Therefore, we have

$$\begin{aligned}
\mathbb{E}\left[\sum_{i=1}^{t} \check{r}_i\right] &\leq \mathbb{E}\left[\sum_{i=1}^{t} \beta_i^{1/2} \sigma_{i-1}(\boldsymbol{x}_i)\right] \\
&\leq \mathbb{E}\left[\left(\sum_{i=1}^{t} \beta_i\right)^{1/2} \left(\sum_{i=1}^{t} \sigma_{i-1}^2(\boldsymbol{x}_i)\right)^{1/2}\right] \\
&\leq \sqrt{\mathbb{E}\left[\sum_{i=1}^{t} \beta_i\right]} \sqrt{\mathbb{E}\left[\sum_{i=1}^{t} \sigma_{i-1}^2(\boldsymbol{x}_i)\right]} \\
&\leq \sqrt{t\mathbb{E}[\beta_t]} \sqrt{\mathbb{E}\left[\sum_{i=1}^{t} \sigma_{i-1}^2(\boldsymbol{x}_i)\right]} \\
&\leq \sqrt{t(2 + 2d\log(\lceil bdrt^2(\sqrt{\log(ad)} + \sqrt{\pi}/2)\rceil))} \sqrt{\mathbb{E}\left[\check{C}_1 \gamma_t\right]} \\
&= \sqrt{\check{C}_1 t \gamma_t (2 + s_t)}.
\end{aligned}$$

$\square$

## B.6 Proof of Theorem A.4

*Proof.* Theorem A.4 is proved by using the same argument as in the proof of Lemma B.1. $\square$

