# OpenReview forum: "Active Learning for Level Set Estimation Using Randomized Straddle Algorithms"
_TMLR — Accepted by TMLR_

### Review · Reviewer_PU6D · 2024-09-01

**Summary Of Contributions:**

This work focus on the Level Set Estimation problem, where:

Theoretical Contributions:
1. A randomized straddle algorithm is proposed, where the model parameter \beta_t^{1/2} is drawn from a chi-squared distribution instead of fixed in previous algorithms (e.g. Gotovos 2013).
2. It is proved that the expected misclassification rate of the proposed model converge to 0, with a convergence rate only related to noise \sigma in the Gaussian Process model and maximum information gain.

Experimental Contributions:
1. Experimental results are provided with superior performance in most of the setups for synthesized dataset and real world dataset.

**Audience:**

Yes

**Broader Impact Concerns:**

No concerns.

**Claims And Evidence:**

Yes

**Requested Changes:**

I would be happy if the authors can:

1. Clarify the difference between Theorem A.3 (Appendix A.2) and Theorem 2, Srinivas et al 2010. and provide evidence that the novelty is significant compare with Srinivas et al 2010.

2. Provide more discussion of the maximum information gain, especially in the infinite data space \mathcal{X} case.

**Strengths And Weaknesses:**

My research expertise resides mainly in Gaussian Process regression, not familiar with Level Set Estimation. The following comments are based on this manuscript and brief reading of the related works LSE (Gotovos 2013) and GP-UCB (Srinivas et al. 2010) model.

Strengths:

1. The paper is well written.

2. Experimental results looks good, justifies the effectiveness of the model.

Weakness:

1. The overall contribution is limited, because 1) the proposed algorithm is an extension of the LSE (Gotovos 2013) model, with \beta_t drawn from a chi-squared distribution (Theorem 1) instead of a fixed value given the size of the finite data space (Theorem 1, Gotovos 2013) and 2) the proof of Theorem 1 on the infinite set \mathcal{X} case (Theorem A.3, Appendix A.2) looks very similar to Theorem 2, Srinivas et al. 2010, which also tackles the infinite set \mathcal{X} case. Thus the claim that this algorithm "can be applied even when \mathcal{X} is an infinite set" may not be a significant contribution.

2. Some statement are unclear. In definition of the maximum information gain \gamma_t (Page 6), the maximization is with respect to what? A subset of \mathcal{X} with t elements? What if \mathcal{X} is continuous as discussed in the Theorem 1?

---

> ### Author Response · Authors · 2024-09-03
> **Answer to Weakness 1**
>
> We thank the reviewer for the constructive suggestions and comments.
>
> **Answer to Weakness 1:**
> ```
> 1. The overall contribution is limited, because 1) the proposed algorithm is an extension of the LSE (Gotovos 2013) model, with \beta_t drawn from a chi-squared distribution (Theorem 1) instead of a fixed value given the size of the finite data space (Theorem 1, Gotovos 2013) and 2) the proof of Theorem 1 on the infinite set \mathcal{X} case (Theorem A.3, Appendix A.2) looks very similar to Theorem 2, Srinivas et al. 2010, which also tackles the infinite set \mathcal{X} case. Thus the claim that this algorithm "can be applied even when \mathcal{X} is an infinite set" may not be a significant contribution.
> ```
> First, regarding 1), Gotovos et al. (2013) performs an operation of taking the intersection of confidence intervals of the black-box function values ​​at each input point for each time (line 6 of Algorithm 1 in Gotovos et al. (2013)), while our method does not perform this operation. Therefore, our approach does not simply replace the confidence parameters of their method with random samples from a chi-squared distribution. This is the difference in the algorithm. Also, the goodness of the LSE algorithm proposed by them is evaluated by the maximum value (max loss) of the deviation from the threshold in the case of a misclassification, while the goodness of our algorithm is evaluated by the average value (mean loss) of the deviation from the threshold in the case of a misclassification. This is the difference in the performance evaluation indexes. Furthermore, Gotovos et al. (2013) guarantees that the max loss is less than or equal to $\epsilon$ with probability at least $1-\delta$. On the other hand, we guarantee that the expected value of the mean loss is less than or equal to $\sqrt{\frac{C_1 \gamma_t}{t}}$. Therefore, the difference is whether the probabilistic upper bound or upper bound of expected values. This is the difference of upper bounds for evaluation indexes. In summary, there are differences in the algorithms, the evaluation indexes, and upper bounds for evaluation indexes. Therefore, we believe that the proposed algorithm is not a simple extension of Gotovos et al. (2013).
>
> Next, regarding 2), Theorem A.3 does not deal with the infinite set case of Theorem 4.1. Theorem 4.1 includes both the cases where $\mathcal{X}$ is finite and infinite. This implies that the proposed method does not require discretization even in the infinite set setting, and furthermore, achieves the same upper bound as in the finite setting. On the other hand, Srinivas et al. (2010) requires discretization when dealing with infinite sets, which increases the number of unknown tuning parameters. In this sense, we believe that the claim in the proposed method that "can be applied even when \mathcal{X} is an infinite set" is a major contribution. In addition, regarding the similarities between the proof of Theorem A.3 and the proof of Theorem 2 in Srinivas et al. (2010), there are two clear differences: i) To begin with, Srinivas et al. (2010) deals with maximization problems, and the quality of the algorithm is based on the regret, the difference between the maximum value and the function value at the observation point at each time. On the other hand, our study deals with LSE, and in Appendix A.2 the quality of the algorithm is based on $\tilde{r}_t$ (page 14), which is essentially different from the regret in maximization problems. ii) As mentioned in the answer to 1), we provide upper bounds for the expected value of the evaluation index of the algorithm. On the other hand, the (cumulative) regret upper bound of Srinivas et al. (2010) is guaranteed probabilistically. In this sense, the results of our method and their results are different. Therefore, although it is true that the upper bounds are similar, the process of arriving at them is essentially different.
>
> Based on the above, we believe that the contribution of the proposed method is not limited. These explanations will be appropriately reflected in the revised manuscript.

---

> ### Author Response · Authors · 2024-09-03
> **Answer to Weakness 2**
>
> **Answer to Weakness 2:**
> ```
> Some statement are unclear. In definition of the maximum information gain \gamma_t (Page 6), the maximization is with respect to what? A subset of \mathcal{X} with t elements? What if \mathcal{X} is continuous as discussed in the Theorem 1?
> ```
> As pointed out by the reviewer, the max operator was inappropriate when $\mathcal{X}$ is an infinite set (continuous). The correct definition is
> \begin{equation}
> \gamma_t = \sup_{ \\{ \tilde{ \boldsymbol{x}}_1,\ldots, \tilde{ \boldsymbol{x}}_t  \\}  \subset \mathcal{X} } \frac{1}{2} \log \det (\boldsymbol{I}_t+\sigma^{-2}\_{{\rm noise}} \tilde{\boldsymbol{K}}\_t). \quad \quad (1)
> \end{equation}
>
> Note that at each time $t \geq 1$, the sup operator is taken for any finite subset of $\mathcal{X}$ with the number of elements is $t$. In the revised manuscript, we will change the definition of $\gamma_t$ to (1). On the other hand, in terms of the definition, it may be appropriate to call it the supremum information gain, but since it is called the maximum information gain even when the definition (1) is used in the infinite setting, such as in [1] below, we will continue to call it the maximum information gain after explaining these explanations.
>
> [1] Delayed Feedback in Kernel Bandits (ICML2023)
>
> In Theorem 1, $\gamma_t $ is used in the sense of (1), regardless of whether $\mathcal{X}$ is finite or infinite.

---

> ### Author Response · Authors · 2024-09-03
> **Answer to Requested Changes**
>
> ```
> 1. Clarify the difference between Theorem A.3 (Appendix A.2) and Theorem 2, Srinivas et al 2010. and provide evidence that the novelty is significant compare with Srinivas et al 2010.
> 2. Provide more discussion of the maximum information gain, especially in the infinite data space \mathcal{X} case.
> ```
> See the answers to Weakness 1 and 2.
> As the reviewer pointed out, since there were indeed some unclear definitions and difficult to understand points, our responses will be appropriately reflected in the revised version.

---

### Review · Reviewer_2yzN · 2024-10-05

**Summary Of Contributions:**

The paper makes advances in level set estimation (LSE), a variant of Bayesian optimization where the main problem is to identify the set of input points where a function takes a value above or below a given threshold $\theta$. Several works have studied and introduced new models/versions around LSE, particularly since GP-UCB (Srinivas, 2010), but one challenge remains: the selection of the confidence parameter $\beta_t^{1/2}$. This parameter basically adjusts the trade-off between exploitation-exploration, making the process more expensive if it is oddly tuned. To solve this problem, the paper introduces a new acquisition function (AF) method where the confidence parameter is sampled independently of the number of iteration, what used to increase it in the past. The main result is that now, having the confidence parameter model by a chi-squared distribution allows a theoretical result around the loss values (with two inequalities in Theorems 4.1 and 4.2), avoids an increasing parameter with the number of iterations and provides equal-or-better performance results than the other 4 or 5 previous SOTA methods

**Audience:**

Yes

**Claims And Evidence:**

Yes

**Requested Changes:**

The manuscript is in general in a good direction. I would love to see some improvements in terms of the weaknesses that I previously mentioned. In general, there is room in the manuscript for further improvement, particularly in the second half -- where theoretical analysis and experiments could even shine a bit more.

**Strengths And Weaknesses:**

### Strengths:

- Introduction of the problem and related work are extremely well written and particularly well fitted for a journal submission like TMLR where clarity and concise references are a must.

- The technical side of the paper, that finally leads to Eq. (3) is understandable and clear. Also the theoretical results in Section 4 are clear and easy to follow for the reader. In general, I find that both the LSE problem is well explained and is clear what the contributions are.

- Even if experimental results are somehow synthetic in most of its parts, the is performance clearly equal-or-better than the rest of SOTA methods. I also appreciate the effort of the authors for making a comparison with 4 or 5 methods that are indeed similar to the one proposed.

### Weaknesses:

- Even if the LSE problem is not exactly BO, I am a bit suprised that little connection are drawn in the manuscript after the related work. For example, the exploration-explotation side of the problem is kind of missed in the methodology. I am saying this, for instance, because it would nice to see what are the consequences in terms of computational cost when the parameter $\beta$ is oddly set and the function of interest is super difficult to evaluate.

- Even if we are treating here an LSE problem, where we are concerned with having function values over or below a certain threshold, I am missing a bit of some visually clear experiments where the effect of a well-fitted confidence parameter shines. Figures are fine, but some other types of results would have helped me to understand better the differences between SOTA methods and this one.

- The role of the loss in pp. 6 and Theorems is clear, but somehow secondary to me. LSE is concerned about a similar BO problem where we want to estimate function values over or below a threshold, fine, but then what does this loss function for us in the problem setting and what type of behavior should we expect? To select subsets of X that target areas with low/higher function values around the threshold, ok, but it is not well explained or at least integrated in the story telling of the paper...

- I assume we are treating here with regression/real-valued problems in terms of f(x), right? No comments are made on this side.

- Experiments are fine, but it would be nicer to test the method in less synthetic problems and perhaps more real-world datasets. Introducing an analysis on the effect of the dimensionality $d$ of X would also help to see if the method also outperforms when large-dimensionality plays a role.

---

> ### Author Response · Authors · 2024-10-11
> **Answer to Weakness 1**
>
> We thank the reviewer for the constructive suggestions and comments.
>
> **Answer to Weakness 1:**
> ```
> Even if the LSE problem is not exactly BO, I am a bit suprised that little connection are drawn in the manuscript after the related work. For example, the exploration-explotation side of the problem is kind of missed in the methodology. I am saying this, for instance, because it would nice to see what are the consequences in terms of computational cost when the parameter
>  is oddly set and the function of interest is super difficult to evaluate.
> ```
>
> For an expensive-to-evaluate black-box function $f(x)$, the objectives of LSE and BO are clearly different. While LSE aims to identify the region where $f(x)$ is above or below a given threshold, BO aims to identify the maximum (or minimum) value of $f(x)$. However, both settings have in common the point that they build a predictive model such as a Gaussian process model for an unknown black-box function $f(x)$ and determine the next evaluation point $x_{next}$ using an acquisition function such as Straddle or UCB based on the predictive model. In the revised version, we will add an explanation that although the objectives of LSE and BO are clearly different, the process of adaptively determining the next evaluation point using an acquisition function based on a predictive model is common. In addition, we will add a statement that since the objective of BO is essentially different from that of LSE, we refer to the survey paper by Shahriari et al. (2015) for a detailed explanation of BO.
>
> From the perspective of exploration-exploitation, when $\beta^{1/2}$ is set too small, only points whose predicted mean is close to the threshold are selected as the next evaluation point, whereas when $\beta^{1/2}$ is set too large, all points tend to be selected almost equally.
>
> Regarding computational cost, in the framework of BO and LSE for black-box functions, "computational cost" is often used to mean the time required to calculate the acquisition function. In this sense, the computational cost is small because it does not depend on $f(x)$ and is easy to calculate regardless of the value of $\beta^{1/2}$, according to the definition of the proposed acquisition function. On the other hand, if the term "computational cost" is used to mean (i) the time required to evaluate the function $f(x)$, or (ii) the number of observations of $f(x)$ required to achieve the desired objective, then (i) does not depend on the value of $\beta^{1/2}$ because the evaluation of the function $f(x)$ is unrelated to the algorithm, but depends on the time required to evaluate $f(x)$. Regarding (ii), when $\beta^{1/2}$ is extremely small or extremely large, the method may fail to obtain necessary points or observe unnecessary points, whereas the proposed method can avoid such behavior.

---

> ### Author Response · Authors · 2024-10-11
> **Answer to Weakness 2**
>
> **Answer to Weakness 2:**
> ```
> Even if we are treating here an LSE problem, where we are concerned with having function values over or below a certain threshold, I am missing a bit of some visually clear experiments where the effect of a well-fitted confidence parameter shines. Figures are fine, but some other types of results would have helped me to understand better the differences between SOTA methods and this one.
> ```
>
> For example, consider $f: [-8,8] \to \mathbb{R}$, a function with peaks near $x=-5$ and $x=5$, and a valley near $x=0$. If $f(x)$ is observed at $x=-5$ as the initial point, setting a small value such as $\beta^{1/2}= 1$ will only observe the inputs around $x = -5$ whose function value is close to the threshold. On the other hand, setting an extremely large value such as $\beta^{1/2}= 10$ will result in behavior in which all points are selected at approximately equal intervals. This is because Straddle is defined as $str(x) = \beta^{1/2} \sigma_t (x) - |\mu_t(x)-\theta|$, and when $\beta^{1/2}$ is small, the second term $- |\mu_t (x)-\theta|$ becomes dominant. In this case, only input points with predicted mean $\mu_t(x)$ near the threshold will be selected, and conversely, when $\beta^{1/2}$ is large, the first term $\beta^{1/2} \sigma_t(x)$ will dominate, and input points with large $\sigma_t(x)$ will tend to be selected, resulting in a behavior that is almost the same as uncertainty sampling, that is, all points are taken almost equally. On the other hand, the proposed method can avoid this behavior. In the revised version, we plan to explain the above with diagrams when explaining the proposal acquisition function.

---

> ### Author Response · Authors · 2024-10-11
> **Answer to Weakness 3**
>
> **Answer to Weakness 3:**
> ```
> The role of the loss in pp. 6 and Theorems is clear, but somehow secondary to me. LSE is concerned about a similar BO problem where we want to estimate function values over or below a threshold, fine, but then what does this loss function for us in the problem setting and what type of behavior should we expect? To select subsets of X that target areas with low/higher function values around the threshold, ok, but it is not well explained or at least integrated in the story telling of the paper...
> ```
>
> The loss function $l_t (x)$ returns the absolute value of the difference between the true value of $f(x)$ at $x$ and the threshold $\theta$ if the classification is incorrect, and the loss is 0 if the classification is correct. For example, the two-dimensional coordinate of a silicon ingot is $x$, and the lifetime value at that coordinate is $f(x) \in [0,10]$ (the larger the value of $f(x)$, the better the performance of the solar cell). In this case, for example, if the threshold is set to $\theta = 3$, we want to accurately determine whether $f(x)$ is above or below $\theta$ at each $x$. Here, if point $x$, where $f(x) = 4$, is classified as the lower side, $l_t (x)= 4-3 = 1$.
> Similarly, if point $x^\prime$, where $f(x^\prime) = 1$, is mistakenly classified as the upper side, $l_t(x^\prime) = 3-1 = 2$. Also, if points $x$ and $x^\prime$ where $f(x)=3.01$ and $f(x^\prime)=10$ are mistakenly classified as the lower side, the latter should be avoided more (it is not good to assume that a point with a very good carrier lifetime value of 10 is the lower side (i.e., a defective area)).  In this case, when the losses of each are calculated, $3.01-3=0.01$ and $10-3=7$ are obtained, and the latter has a larger loss, which is consistent with reality.
>
> Also, the loss $r(H_t,L_t)$ is the average of the loss $l_t(x)$ at each point $x$. For example, in an actual data experiment on a silicon ingot, the coordinates of more than 6,000 points are considered, and if the loss $l_t (x)$ is large only at a few points, the average $r(H_t,L_t)$ is small, and it can be seen that it is not a bad classification when viewed from the entire ingot surface. On the other hand, if the value of $r(H_t,L_t)$ is poor, that is, if most of the ingot surface is misclassified, it suggests that the classification is not very accurate.
>
> Since $l_t(x)$ is an index that becomes 0 if classification is correct, and $r(H_t,L_t)$ is an index that becomes 0 if classification is correct for all points, we want $l_t(x)$ and $r(H_t,L_t)$ to become small from early iterations, ideally to become 0.

---

> ### Author Response · Authors · 2024-10-11
> **Answer to Weakness 4**
>
> **Answer to Weakness 4:**
> ```
> I assume we are treating here with regression/real-valued problems in terms of f(x), right? No comments are made on this side.
> ```
>
> In this paper, we consider LSE for $f(x)$, not regression. The function $f(x)$ is the relationship between the input $x$ and the corresponding value. For example, in the silicon ingot example explained in the introduction, $x$ represents the two-dimensional coordinate of the surface of the silicon ingot, and $f(x)$ is the lifetime value at that coordinate (a value that represents the performance of the ingot, which is actually measured by X-ray irradiation, etc., and is not regressed). In this way, $f(x)$ itself is not related to regression.
>
> For theoretical validity, this paper assumes that $f(x)$ is a sample path from a Gaussian process with kernel $k(\cdot,\cdot)$. However, a Gaussian process regression model is used as a model to predict $f(x)$. Nevertheless, the purpose of this paper is LSE for $f(x)$, not regression. Also, since $f:\mathcal{X} \to \mathbb{R}$, $f(x)$ is a real-valued function.

---

> ### Author Response · Authors · 2024-10-11
> **Answer to Weakness 5**
>
> **Answer to Weakness 5:**
> ```
> Experiments are fine, but it would be nicer to test the method in less synthetic problems and perhaps more real-world datasets. Introducing an analysis on the effect of the dimensionality
>  of X would also help to see if the method also outperforms when large-dimensionality plays a role.
> ```
>
> First, the main purpose of this method is not to mention that it dramatically improves practical performance, but to emphasize that it is an easy-to-use method that does not require parameter adjustment. In the experimental part, we deal with a theoretically ideal setting (GP sample path), a theoretically non-ideal but easy-to-treat setting (two discrete benchmark function settings), a difficult-to-treat setting (three continuous benchmark settings), and one real data. In this sense, we consider this to be sufficient because there are six settings that are not guaranteed to work theoretically, and we also deal with real data.
>
> Regarding the increase in the number of dimensions, it is an interesting point of view, but we do not consider it to be a topic that will be addressed in this paper. In fact, high-dimensional BO is a topic that should be discussed on its own, and we would like to add the high-dimensional extension of this method to our future work.

---

> ### Author Response · Authors · 2024-10-11
> **Answer to Requested Changes**
>
> ```
> The manuscript is in general in a good direction. I would love to see some improvements in terms of the weaknesses that I previously mentioned. In general, there is room in the manuscript for further improvement, particularly in the second half -- where theoretical analysis and experiments could even shine a bit more.
> ```
>
> See the answers to weakness comments.

---

> > ### Comment · Reviewer_2yzN · 2024-11-15
> > **Post-rebuttal comments**
> >
> > I would like to thank the authors for their thorough responses to all my concerns and for incorporating the feedback I gave into the manuscript with new updates of the unclear (or less clear) parts. I've already delivered my recommendation of the manuscript, which is positive, in general.
> >
> > Best,
> > R2yzN

---

### Review · Reviewer_ok7A · 2024-10-11

**Summary Of Contributions:**

This paper introduces the Randomized Straddle Algorithm for Level Set Estimation (LSE) in black-box functions. The proposed method offers theoretical guarantees and eliminates the need for any user-specified parameters to be adjusted. The effectiveness of this approach is validated through extensive numerical experiments conducted on both synthetic and real-world data.

**Audience:**

Yes

**Broader Impact Concerns:**

No statement-requiring ethical implication from my point of view.

**Claims And Evidence:**

Yes

**Requested Changes:**

1.	A chi-squared distribution with 2 degrees of freedom (k = 2) is an exponential distribution with a mean value of 2.  To maintain consistency with previous works, I suggest that the author use the description of the exponential distribution in the manuscript.

2.	Although [2] is more solid than [1], I still recommend that the authors also cite [1], even if it serves as a stepping stone.

3.	The authors should list the drawback and improvements of the proposed algorithms more in detail.

4.	Page 5, ‘In contrast, the random sample proposed in this study does not require the addition of such a constant’ Is the random sampling approach in this paper significantly different from the random sampling used in reference [2]? Can the techniques presented here be applied to the problems studied in [2]? In other words, is the random sampling method proposed in this paper a general technique that can be adapted for other applications?

[1] Berk, Julian, Sunil Gupta, Santu Rana, and Svetha Venkatesh. "Randomised Gaussian process upper confidence bound for Bayesian optimisation." In Proceedings of the Twenty-Ninth International Conference on International Joint Conferences on Artificial Intelligence, pp. 2284-2290. 2021.

[2] Takeno, Shion, Yu Inatsu, and Masayuki Karasuyama. "Randomized Gaussian process upper confidence bound with tighter Bayesian regret bounds." In International Conference on Machine Learning, pp. 33490-33515. PMLR, 2023.

**Strengths And Weaknesses:**

**Strengths**:

1.	The presentation of the work is clear.

2.	The proposed method is simple yet effective.

3.	The proposed method provides strong theoretical guarantees regarding the expected loss for misclassification.

**Weaknesses**:

1.	The idea of replacing user-specified value of $\beta$ in UCB by a random sample from a distribution is not new [1,2]. For readers familiar with references [1] and [2], it seems that they can grasp the main conclusions of this article based on the abstract alone. Then, what’s the main challenge when extending this idea to level set estimation? Additionally, are there any new insights when extending this idea to level set estimation? It seems such idea can also be easily extended for other scenarios like multifidelity BO, multiobjective BO, high-dimensional BO, and robust BO.

2.	Determining the value of $\beta$ through random sampling is helpful for analyzing the average-case regret. In experimental studies, the author repeated each experiment 100 times. However, in real-world scenarios, users typically have only one opportunity to run the algorithm. Then, are there any regret analysis results on the high probability bound?

3.	Throughout the paper, the analysis on the drawback and limitations of determining the value of $\beta$ through random sampling is deficient.

[1] Berk, Julian, Sunil Gupta, Santu Rana, and Svetha Venkatesh. "Randomised Gaussian process upper confidence bound for Bayesian optimisation." In Proceedings of the Twenty-Ninth International Conference on International Joint Conferences on Artificial Intelligence, pp. 2284-2290. 2021.

[2] Takeno, Shion, Yu Inatsu, and Masayuki Karasuyama. "Randomized Gaussian process upper confidence bound with tighter Bayesian regret bounds." In International Conference on Machine Learning, pp. 33490-33515. PMLR, 2023.

---

> ### Author Response · Authors · 2024-10-15
> **Answer to Weakness 1**
>
> We thank the reviewer for the constructive suggestions and comments.
>
> **Answer to Weakness 1:**
> ```
> The idea of replacing user-specified value of in UCB by a random sample from a distribution is not new [1,2]. For readers familiar with references [1] and [2], it seems that they can grasp the main conclusions of this article based on the abstract alone. Then, what’s the main challenge when extending this idea to level set estimation? Additionally, are there any new insights when extending this idea to level set estimation? It seems such idea can also be easily extended for other scenarios like multifidelity BO, multiobjective BO, high-dimensional BO, and robust BO.
> ```
>
> Indeed, the proposed method is similar to existing methods in that the confidence parameter in the acquisition function is randomly sampled, but there are some points that require careful consideration when performing theoretical analysis in the LSE setting. In the first place, Takeno et al. (2023) treat a maximization problem and consider the regret $r_t = f(x^\ast) - f(x_t)$, where $x^\ast$ is the maximum solution. As an evaluation index for theoretical analysis, they consider a Bayesian cumulative (or simple) regret, but the theoretical validity of their method is based on the fact that $f(x^\ast)$ can be bounded from above with high probability, and as a result, the expected value of $f(x^\ast)$ can be bounded at a certain expected value (Lemma 4.1 and 4.2 in Takeno et al. (2023)), which is achieved by using UCB. On the other hand, since the losses $l_t(x)$ and $r_t(H_t,L_t)$ in LSE dealt with in this paper are essentially different from the regret $r_t$ and Bayesian cumulative (or simple) regret in maximization problems, it is not obvious that claims similar to Lemma 4.1 and 4.2 in Takeno et al. (2023) can be derived simply by randomizing the confidence parameter of some acquisition function in LSE.
>
> In addition, in existing LSE studies such as Gotovos et al. (2013), the classification rules include the posterior mean, posterior standard deviation, and $\beta_t$, and the loss function depends on the classification rule. Therefore, since the classification rules include $\beta_t$, randomized analysis was difficult. On the other hand, the loss in the proposed method is based on the classification rules, but is different from theirs in the first place. Furthermore, since our classification rules use only the posterior mean, the classification rules themselves do not include $\beta_t$, which makes randomized analysis possible.
>
> In this paper, we have clarified that theoretical analysis can be performed by considering a straddle acquisition function and appropriately designing the distribution of the confidence parameter. In other words, we have derived non-trivial theoretical analysis results that can only be derived by solving the problem of considering an appropriate acquisition function for an appropriate loss function and correctly designing the distribution of the confidence parameter, and we believe that our contribution is significant and is not a simple extension of existing methods.
>
> The insight gained from the derivation of the theoretical analysis results in this paper is that even if the problem setting is different and the evaluation index considered changes, if an acquisition function is designed that can bound the evaluation index with  high probability, it can be expected to be extended to other problem settings. In particular, both the results of Takeno et al. (2023) and the results of this paper use the property that some loss function can be bounded from above using $\beta^{1/2}_t \sigma _{t-1} (x_t)$ with high probability, so it can be said that the key point of the theoretical analysis is to consider a combination of acquisition function and loss that satisfies this property.
>
> Regarding extension to other problem settings, it is necessary to carefully consider the loss considered and the corresponding appropriate acquisition function and distribution of the confidence parameter, so it cannot be simply extended using the contents of this paper.

---

> ### Author Response · Authors · 2024-10-15
> **Answer to Weakness 2**
>
> **Answer to Weakness 2:**
> ```
> Determining the value of through random sampling is helpful for analyzing the average-case regret. In experimental studies, the author repeated each experiment 100 times. However, in real-world scenarios, users typically have only one opportunity to run the algorithm. Then, are there any regret analysis results on the high probability bound?
> ```
>
> By considering Markov's inequality $P(X \geq a) \leq \frac{E[X]}{a}$, $a>0$, we can derive a high probability bound. In fact, if we consider $r(H_t,L_t)$ as $X$, then by Theorem 4.2, the following inequality holds:
> $$
> P(r(H_t,L_t) \geq a) \leq \frac{E[r(H_t,L_t)]}{a} \leq \frac{1}{a} \sqrt{\frac{C_1 \gamma_t}{t}}.
> $$
> Therefore, if we set $a= \delta^{-1} \sqrt{\frac{C_1 \gamma_t}{t}}$, then for each $t$, $r(H_t,L_t) \leq \delta^{-1} \sqrt{\frac{C_1 \gamma_t}{t}}$ holds with probability at least $1-\delta$. However, since the right-hand side contains a term of $\delta^{-1}$, the challenge is to improve it to the order of $O(\sqrt{ \log (\delta^{-1}) })$, as in Theorem 1 of Srinivas et al. (2010).

---

> ### Author Response · Authors · 2024-10-15
> **Answer to Weakness 3**
>
> **Answer to Weakness 3:**
> ```
> Throughout the paper, the analysis on the drawback and limitations of determining the value of through random sampling is deficient.
> ```
>
> Compared to Straddle (Bryan et al. (2005)) alone, we believe there is no drawback to randomly sampling $\beta$, because when $\beta$ is determined by the user, there is no theoretical guarantee if practical performance is emphasized, and practical performance deteriorates if theoretical guarantee is emphasized, while random sampling has been demonstrated to achieve both practical performance and theoretical guarantee.
>
> On the other hand, compared to existing methods such as MILE, the drawbacks and limitations of the proposed method are practical performance and scalability. The proposed method does not have any significant changes other than adding randomization to the existing straddle, so practical performance is not dramatically improved compared to when a fixed $\beta$ is used. For example, if we can design an acquisition function that sets the next point $x_{next}$ to the point that most improves $E[r(H_t,L_t)]$, we can expect practical performance to improve, but theoretical analysis is not easy. This is one of the drawbacks and limitations.
>
> In addition, the theoretical analysis of the proposed method was obtained by carefully designing the loss, acquisition function, and $\beta$ distribution. Therefore, for example, if the proposed method is used when the F-score is used as an evaluation index, or even if the $\beta$ of the MILE acquisition function is randomly sampled, there is no guarantee that a similar theoretical analysis can be performed. As a result, the method is highly problem-dependent and difficult to expand. This is also one of the drawbacks and limitations.
>
> Moreover, the fact that the high-probability bound derived from the proposed method is not tight is a drawback. The loss of the proposed method is different from existing losses (see the answer to (1)), and one of the contributions is that we were able to derive the  bound for its expected value, but as shown in the answer to (2), the high-probability bound includes a term of $1/\delta$. On the other hand, many existing studies that perform theoretical analysis of high-probability bounds, such as the UCB algorithm and the LSE algorithm, show that the term $\log(1/\delta)$ appears in the bounds of evaluation index such as cumulative regret and $\epsilon$- accuracy, so one of the drawbacks of the proposed method is that the term $1/\delta$ in the high-probability bounds is not tight.

---

> ### Author Response · Authors · 2024-10-15
> **Answer to Requested Changes**
>
> **Answer to Requested Changes 1:**
> ```
> A chi-squared distribution with 2 degrees of freedom (k = 2) is an exponential distribution with a mean value of 2. To maintain consistency with previous works, I suggest that the author use the description of the exponential distribution in the manuscript.
> ```
> In the revised version, we will explain that the chi-squared distribution with 2 degrees of freedom is the exponential distribution with mean 2 (i.e., parameter 1/2).
>
> ---
>
> **Answer to Requested Changes 2:**
> ```
> Although [2] is more solid than [1], I still recommend that the authors also cite [1], even if it serves as a stepping stone.
> ```
> In the revised version, we will add [1] to the citation in addition to [2].
>
> ---
> **Answer to Requested Changes 3:**
>
> ```
> The authors should list the drawback and improvements of the proposed algorithms more in detail.
> ```
> In the revised version, based on the response to the Weakness comment 3, we will explain in more detail the drawbacks and limitations of the proposed method.
>
> ---
> **Answer to Requested Changes 4:**
>
> ```
> Page 5, ‘In contrast, the random sample proposed in this study does not require the addition of such a constant’ Is the random sampling approach in this paper significantly different from the random sampling used in reference [2]? Can the techniques presented here be applied to the problems studied in [2]? In other words, is the random sampling method proposed in this paper a general technique that can be adapted for other applications?
> ```
>
> As stated in the answer to Weakness Comment 1, the only thing our method has in common with existing methods is that $\beta$ is randomly sampled, and it clearly differs from existing methods in that when performing theoretical analysis, the evaluation index, acquisition function, and distribution of $\beta$ must be properly designed. Therefore, simply extending this method will not allow it to be applied to other settings. In the revised version, we will add this explanation to the limitations.

---

> > ### Comment · Reviewer_ok7A · 2024-10-15
> > **Comments to Author's Point-by-Point Reply**
> >
> > I have reviewed the comments from the other reviewers as well as the authors' responses. I am satisfied with the authors' replies and have no further questions.

---

### Comment · Action_Editor_VTJX · 2024-10-03
**Update of the Current Review Process**

Dear Authors,

Thank you for submitting your work to TMLR, and sorry for the long waiting time after the review deadline. Here is the update of the current review process.

A new review comment is expected to be available this weekend by one of the original reviewers. In addition, a new emergent reviewer has been recruited, and the review comment will be available by next Friday. Once three review comments are obtained, an unresponsive original reviewer will be removed from this work.

Best Regards,

AE

---

### Author Response · Authors · 2024-10-21
**Revised Manuscript**

We submitted the revised manuscript that appropriately incorporated the reviewers' comments. Revisions in the manuscript are written in red. The changes are as follows:

[1] Citation of the RGP-UCB paper (Reviewer ok7A): On page 2, lines 3-8, we have cited the RGP-UCB paper and explained RGP-UCB.

[2] Relation between BO and LSE (Reviewer 2yzN): On page 2, footnote, we have added the explanation of the relation between BO and LSE.

[3] Chi-squared distribution and exponential distribution (Reviewer ok7A): On page 3, lines 15-16, we have added the description that the chi-squared distribution with two degrees of freedom is equal to the one-parameter exponential distribution with parameter 1/2.

[4] Behavior of the proposed acquisition function with different $\beta_t$ (Reviewer 2yzN): On page 5, lines 29-30, and Figure 2 on page 6, we have added the graphical illustration of how difference in $\beta_t$ affect the selected points.

[5] Definition of the maximum information gain (Reviewer PU6D): On page 6, lines 20-21 and footnote, we have added the correct definition of the maximum information gain and its explanations.

[6] Theorem 4.1 and 4.2 hold whether $\mathcal{X}$ is finite or infinite (Reviewer PU6D): On page 7, line 6, we have emphasized that Theorem 4.1 and 4.2 hold whether $\mathcal{X}$ is finite or infinite.

[7] Comparison with existing methods (Reviewer ok7A, PU6D): On page 7, lines 11-31, we have added the explanations that the theoretical results of the proposed method cannot be derived by simply randomizing existing methods including the difference between the proposed and existing methods.

[8] High-probability bound (Reviewer ok7A): From page 7, line 40 to page 8, line 8, we have added the theoretical result on high-probability bounds.

[9] Explanation of non-trivial theoretical results (Reviewer ok7A, PU6D): On page 12, lines 10-13, we have added the explanation that the theoretical result in the proposed method was non-trivial and derived by appropriately designing the loss, acquisition function and distribution of the confidence parameter.

[10] Drawbacks and limitations (Reviewer ok7A): From page 12, line 25 to page 13, line 6, we have added the explanation of three drawbacks and limitations in the proposed method.

[11] Extension to high-dimensional settings (Reviewer 2yzN): On page 13,lines 7-9, we have added that the high-dimensional extension is one of future work.

---

> ### Author Response · Authors · 2024-10-25
> **Revised Manuscript Version 2**
>
> On page 6, line 21, we have changed the statement "... where {$ \tilde{x}_1, \ldots , \tilde{x}_t  $} is an arbitrary subset of $\mathcal{X}$ with the number of elements is $t$, ..."  in the revised manuscript to "... where $ \tilde{x}_1, \ldots , \tilde{x}_t  $ are any elements of $\mathcal{X}$, ..." in the latest manuscript because $\tilde{x}_1,\ldots , \tilde{x}_t$ can have the same points and {$ \tilde{x}_1, \ldots , \tilde{x}_t  $} is not necessarily a subset of the number of elements $t$.

---

### Decision · Action_Editor_VTJX · 2024-11-16

**Recommendation:** Accept as is

**Comment:**

The reviewers find this work is clear and well written, the proposed method is effective, and the experimental results are good. They also appreciate the clear, easy-to-follow, and strong theoretical analysis provided in this work. After rebuttal, most of the concerns raised were properly addressed, and one reviewer voted to accept this paper while two reviewers leaned toward accepting. In the official recommendation, Reviewer 2yzN suggests that the high-dimensional side of the problem could be worthy to be explored in future follow-up work, which I also believe is a very good suggestion.

I read the paper in detail and totally agree with all reviewers that this work clearly meets the TMLR acceptance criteria (solid and well-supported claims, potential audience). Therefore, I am happy to accept this work as is.

**Audience:**

All reviewers believe some individuals in TMLR's audience could be interested in the findings of this paper.

**Claims And Evidence:**

This work proposes a novel randomized straddle algorithm for level set estimation (LSE) with black-box functions. The key idea is to use a random confidence parameter $\beta_t$ in the straddle algorithm, which is sampled from the chi-squared distribution with two degrees of freedom. In this way, the confidence parameter does not require adjustment, does not depend on the number of iterations or candidate points, and is not conservative. Thorough theoretical analysis and numerical experiments are provided to show the advantages of the proposed method.

All reviewers believe the claims made in this paper are supported by accurate, convincing and clear evidence.